# Emergence of syntax and word prediction in an artificial neural circuit of the cerebellum

Keiko Ohmae [1,2] & Shogo Ohmae [1,2] ✉

The cerebellum, interconnected with the cerebral neocortex, plays a vital role in human-characteristic cognition such as language processing, however, knowledge about the underlying circuit computation of the cerebellum remains very limited. To gain a better understanding of the computation underlying cerebellar language processing, we developed a biologically constrained cerebellar artificial neural network (cANN) model, which implements the recently identified cerebello-cerebellar recurrent pathway. We found that while cANN acquires prediction of future words, another function of syntactic recognition emerges in the middle layer of the prediction circuit. The recurrent pathway of the cANN was essential for the two language functions, whereas cANN variants with further biological constraints preserved these functions. Considering the uniform structure of cerebellar circuitry across all functional domains, the single-circuit computation, which is the common basis of the two language functions, can be generalized to fundamental cerebellar functions of prediction and grammar-like rule extraction from sequences, that underpin a wide range of cerebellar motor and cognitive functions. This is a pioneering study to understand the circuit computation of human-characteristic cognition using biologically-constrained ANNs.

Language comprehension is a vital cognitive function that supports human communication and knowledge acquisition, however, the underlying mechanisms at the level of neuronal activity within specific brain circuits remain largely unexplored. Artificial neural networks (ANNs) provide an invaluable tool to examine how the brain processes language, because there are no established animal models and only limited data from invasive recordings in humans. While recent AI advancements have led to ANN models that attempt to replicate brain functionality, these models do not adequately account for biological constraints. For instance, neocortical models for language processing based on the transformer algorithm[1–3], which are widely used in language-processing AI like ChatGPT, deviate from actual brain physiology. This is because the transformer-based models lack artificial neurons and process a sentence as a single unit rather than a sequence of individual words. Thus, creating biologically-constrained ANN models is essential to elucidate the computational processes of brain circuits involved in language processing.

Language comprehension after word recognition requires cooperation between the left neocortical language area and the right lateral cerebellum (right Crus I/II)[4–17]. The cerebellum has a large capacity for plasticity, and interestingly, according to recent observations that childhood damage to the right cerebellum causes more permanent and severe language deficits than adult damage[18–21] and that the right cerebellum is activated during learning of new language tasks[16,22,23], cerebellar language acquisition is considered to support subsequent acquisition in the neocortex[15,18,19]. To understand the origins of language functions in the neocortex, it is crucial to understand the language processing mechanisms in the cerebellum at the circuit computation level. Here, we examine how language functions emerge in an ANN model that is biologically-constrained by the known connectivity and inputs/outputs of the cerebellar circuit (cANN).

Sentence comprehension involves recalling the meanings of words and combining them appropriately based on grammar to compose the meaning of the sentence. During this process, two major

[1]Neuroscience Department, Baylor College of Medicine, Houston, TX, USA. [2]Present address: Chinese Institute for Brain Research (CIBR), Beijing, China. ✉e-mail: sohmae.jp@gmail.com

language functions have been linked to the right lateral cerebellum: The first is (1) next-word prediction[4,6–10], which supports faster and more accurate sentence comprehension by placing input words in a predictive context (particularly in noisy environments)[1,6,8,10,24–27]. Although neural activity related to this function has also been reported in neocortical language areas[1,3,24], more substantial evidence including causality has accumulated for the cerebellum[6–9]. The second function is (2) grammatical processing, especially syntactic recognition of subject–verb–object (S-V-O) information[5,11–13,17,28]. These two language functions reflect two more general cerebellar functions that support a variety of cognitive functions. One is prediction of external events[4,5,10,15,29–32], and (1) next-word prediction is a language case of this general function[6–8,10,15,26]. The other is rule extraction from the sequence of events[11,13,33], and (2) syntactic recognition is a language case of this function[11,13,34]. Although the two general functions underlie various cerebellar cognitive functions[4,10,15,29,32] (see Discussion), it remains unclear how these different functions are realized in a cerebellar circuit with a uniform cytoarchitecture. To address this gap in knowledge, elucidation of the underlying network dynamics by ANN is needed[15]. Therefore, in the context of language processing, we tested whether a single circuit architecture can support both general functions, thereby shedding light on the general circuit computation of the cerebellum.

Here, we found that the cANN can acquire prediction in language (i.e., next-word prediction) through training, and simultaneously rule extraction from sequences (i.e., syntactic recognition) can also be acquired spontaneously in the middle layer of the same circuit. This suggests that two general functions of the cerebellum, prediction and rule extraction from sequence, can be captured by a single circuit computation of the cerebellum. This also suggests a potential therapeutic approach for language dysfunction, in which training in word prediction leads to improvement in syntactic comprehension.

## Results

### The cerebellar ANN model as a next-word prediction circuit

We started by assessing whether the biologically-constrained cANN could learn next-word prediction. Our cANN contains three layers—an input layer (granule cells), a middle layer (Purkinje cells), and an output layer (nuclei neurons), and the cANN consists of a conventionally reported feedforward pathway and a recently identified recurrent pathway (brown, blue pathways in Fig. 1b, respectively). The recurrent pathway contains indirect[35–37] and direct output cell-input cell pathways. The direct pathway is known as the nucleocortical pathway[38–44], that has been demonstrated to be essential for prediction in the cerebellum[38,41,45]. The cANN circuit also incorporates the climbing fiber pathway (gray, Fig. 1b) which delivers prediction errors to the cerebellum[6,10,29,46–48]. Based on physiological observations that the prediction signal generated by Purkinje cells after a cue event is persistent until the actual event occurs[38,49–52], we assumed that the prediction signal of Purkinje cells in the cANN is persistent until the next word (see Discussion). This persistent prediction signal maintains the recurrent signal, allowing it to be integrated with the next word. Additionally, the circuit output signal is also maintained and can be compared with the correct answer signal of the actual next word, to compute the prediction error signal.

We designed the inputs/outputs of the cANN (Fig. 1b) according to previous proposals for the next-word prediction circuit of the cerebellum as in Fig. 1a[6–8,10,26]. Based on the sparse coding theory of the input cells (i.e., granule cells), we assumed that words were represented by sparse coding in the input cells: each of the 3000 cells was randomly assigned to one of 3000 words and hence the input signal contained no semantic or grammatical information at all (see below). To facilitate the investigation of information processing in this network, we assumed that the correct answer signal (correct answer = actual next word in Fig. 1b, gray) was encoded by the same 3000-dimensional sparse coding as in the input cells, so that the correct answer signal contained no syntactic information at all (see below). Because the dimensionality of the output cells is theorized to be the same as that of the correct answer signal[29,53–55], we set the number of the output cells as 3000. Through the training of cANN, the activity intensity of each output cell can indicate the likelihood of the corresponding word being the next word (percentages in Fig. 1b). Solely for performance evaluation purposes, we selected the five output cells with the strongest activity and the corresponding five words as the prediction candidates for the next word. Additionally, to replicate the anatomical convergence from Purkinje cells to the output cells (i.e., the number of Purkinje cells > that of the output cells), we created and examined another cANN with this biological convergence (see below).

For learning, the correct information for the next word was delivered to the circuit word by word and the synaptic weights were updated to reduce the prediction error, in line with the learning rule of the cerebellum (Fig. 1b, gray)[10,56,57].

Training the cANN with sentences (e.g., from classic novels) significantly improved the prediction error and the correct prediction rate (Fig. 1c, d). At the very early stages of training, the correct prediction rate jumped from 0% to about 17% (Fig. 1d, gray arrowhead), and at this point the prediction was based on the words that appeared most frequently in the training sentences, regardless of the identity of the immediately preceding word (Supplementary Fig. 1a, b). Then, the correct rate slowly increased to about 40%, where it plateaued; here, the cANN achieved correct prediction rates higher than chance regardless of the frequency of occurrence of the target word (Supplementary Fig. 1f), and the cANN made different next-word predictions after each of the two instances of "the" in a sentence (Fig. 1e, gray boxes), indicating that the circuit utilized words prior to the immediately preceding word.

Next, to replicate the finding from human cerebellar research[7,8], we examined whether the cANN could predict the noun after a verb. For the sentence "The man will sail [verb] the boat [noun]" used in the previous research, together with two similar sentences, the cANN predicted the correct next words and synonyms (Figs. 1e, 2a, b, black boxes). Predicting the noun after a verb had a lower correct rate (median, 22.9%; interquartile range (IQR), 21.0%–23.4%) than the overall prediction rate (38.3%; 37.8–38.6%), possibly because verbs often have a large number of potential noun partners (e.g. partners of "eat"). By contrast, predictions were more accurate for other word types after a verb (for prepositions: 60.0%, 56.7–63.3; object pronouns: 56.5%, 52.2–60.9). The cANN also made proper predictions for other word types, such as nouns after prepositions (43.0%, 42.1–45.6), and adjectives after "be" verbs (27.5%, 25.0–30.0). From these results, we conclude that the cANN is able to function as a next-word prediction circuit, although the correct rate varies with the word type.

To investigate the mechanism of information processing, we examined the role of the Purkinje cells, where the feedforward signal of a newly arriving word and the recurrent signal containing the previous word information are integrated (Fig. 1b). When we visualized the word-by-word transition of Purkinje cells activity in the sentences using principal component analysis (PCA), three different instances of "the" generated different activity patterns in the Purkinje cells (Fig. 2c, the[1], the[2], the[3]). Since the Purkinje cells are the only element of the circuit upstream of the output cells responsible for predictions, these differences are essential for generating different predictions (Fig. 2a, b, gray and black boxes). When the recurrent input was blocked, the activity in the Purkinje cells became identical for all three instances of "the" (Supplementary Fig. 1c), and the impaired cANN no longer produced different predictions (Fig. 2d), indicating that only the immediately preceding word is encoded. More generally, the prediction of the noun after a verb was dramatically degraded in the impaired cANN (correct rate, 2.1%; IQR, 1.1–2.1%).

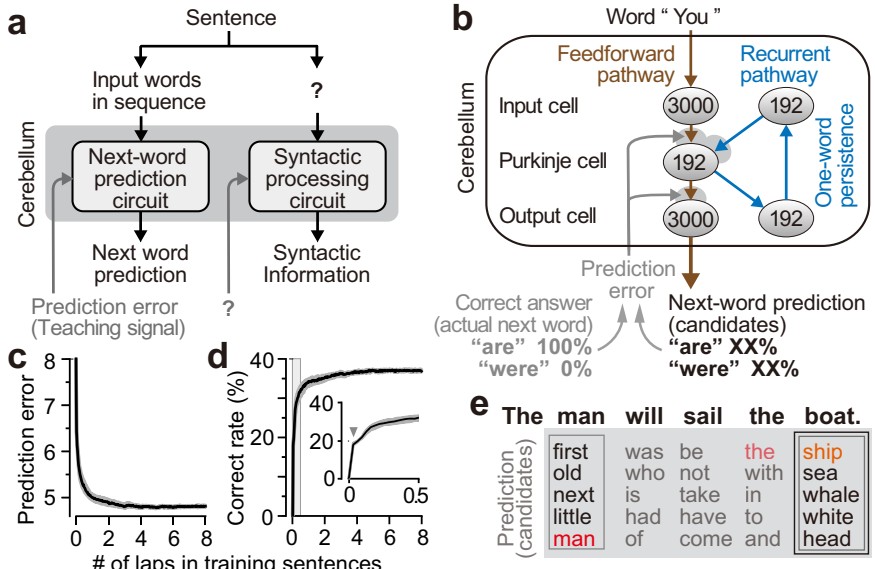

**Fig. 1 | Local circuit model of a cANN to reconstruct cerebellar language functions. a** Previously proposed block diagram with Inputs/outputs for next-word prediction. **b** The three-layer cerebellar circuit, consisting of a feedforward pathway (brown) and a recurrent pathway (blue). The climbing fiber pathway (gray arrows) delivers the prediction error signal to update the synaptic weights on the Purkinje and output cells (gray shading). **c**, **d** Learning curves for the prediction error (**c**; cross entropy; median and IQR) and percentage correct predictions (**d**) for 20 cANNs trained on next-word prediction. Inset, magnified view of the early learning period (gray interval). **e** Predictions (five candidate words) for the two instances of "the" in a sentence by a well-trained cANN. Correct predictions are highlighted in red, and synonyms are in orange.

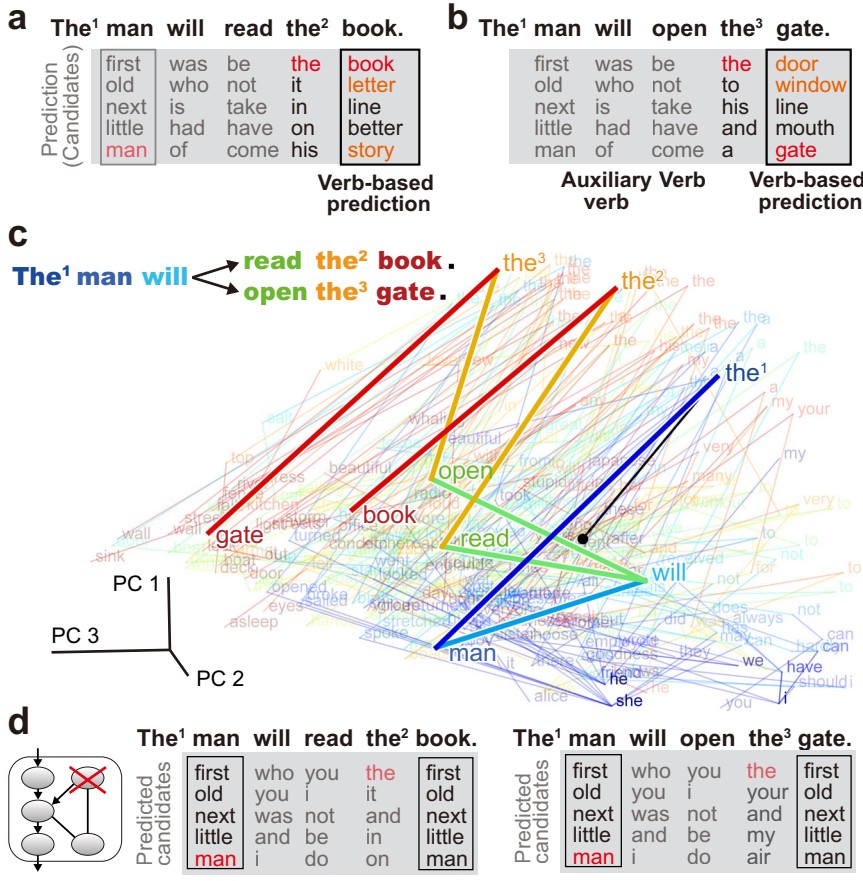

**Fig. 2 | Next-word prediction and network dynamics of a representative cANN.**
**a**, **b** Predictions after three instances of "the" (indexed) in two sentences.
**c** Visualization of the neural activity of Purkinje cells after input of each word by PCA. Words are color-coded in rainbow from the beginning of the sentence to the end, and spontaneous activity is indicated by the black dot. **d** Predictions when the recurrent signal was blocked.

## The cANN model as a syntax-processing circuit

There is little-to-no evidence as to how the inputs/outputs should be designed for another cANN dedicated to syntactic processing[15] (Fig. 1a). For the cerebellum to assist the neocortex in language acquisition, the cerebellum must acquire language processing abilities that the neocortex has not yet acquired. Inspired by the potential of the middle layer of recurrent ANNs[58–60], we investigated whether subject–verb–object (S-V-O) syntactic information could emerge within the next-word prediction circuit of the cANN, whose input signals contain no syntactic information. Indeed, the cANN reliably predicted objective case pronouns after a verb (Supplementary Fig. 1d, e), suggesting that the predictions reflect S-V-O information. Since the Purkinje cells may integrate new and past words according to S-V-O structure, we visualized the activity of the Purkinje cells by PCA (Supplementary Fig. 2a) and by linear support vector machine (Fig. 3a). We found that S words were clustered and well separated from V-O words (blue circle), regardless of the number of words forming the subject (black trace). The separation of S words from V-O words (96.1%; 95.7–96.6; nonlinear support vector machine), V words (96.0%; IQR, 95.7–96.5), and O words (95.8%; 95.4–96.1) was highly accurate (visualized in Supplementary Fig. 2b). Since syntactic information is not included in the correct answer of the next word (confirmed in Fig. 3d, right), these results indicate that the Purkinje cells in the word prediction circuit are capable of extracting highly accurate SVO syntactic information independent of external information. Importantly, even for the same word, the cANN can distinguish when the word appears as a subject (e.g., "the" as a part of a subject, blue arrowhead in Fig. 3a) and when it appears as an object

("the" as a part of object, yellow arrowheads), ensuring that it has S-V-O recognition capability. Thus, to our surprise, syntactic processing does not require a dedicated circuit (Fig. 3b). This may be why the conventional approach of assuming a dedicated circuit as in Fig. 1a worked well for next-word prediction, but not for syntactic processing[15].

To investigate how syntactic information emerges in the cANN, we compared the syntactic information along the feedforward pathway (Fig. 3d). We confirmed that the input cells contain no syntactic information at all, and found that syntactic information was predominantly extracted in the Purkinje cells and was degraded in the output cells (p < 0.001 for any of S, V, and O, Wilcoxon signed-rank test; visualized by distorted distributions in Supplementary Fig. 2c), probably because their direct task is next-word prediction. Furthermore, when the recurrent signal was blocked, the separation of S and O dropped sharply, revealing the importance of the recurrent signal for processing syntactic information (Fig. 3e).

Examining the extent to which S-V-O syntactic information is a primary piece of information in Purkinje cells, we found that the dimensions used for S-V, V-O, and S-O separation by the support vector machine were all similar to the major dimensions of PCA space in terms of direction (Fig. 3c) and information content (Supplementary Fig. 2d, e). This indicates that S-V-O syntactic information is the primary information represented by the activity of Purkinje cells.

### Further biologically-constrained cANN variants

We developed three new cANN variants incorporating further biological constraints and assessed their performance. In the first variant,

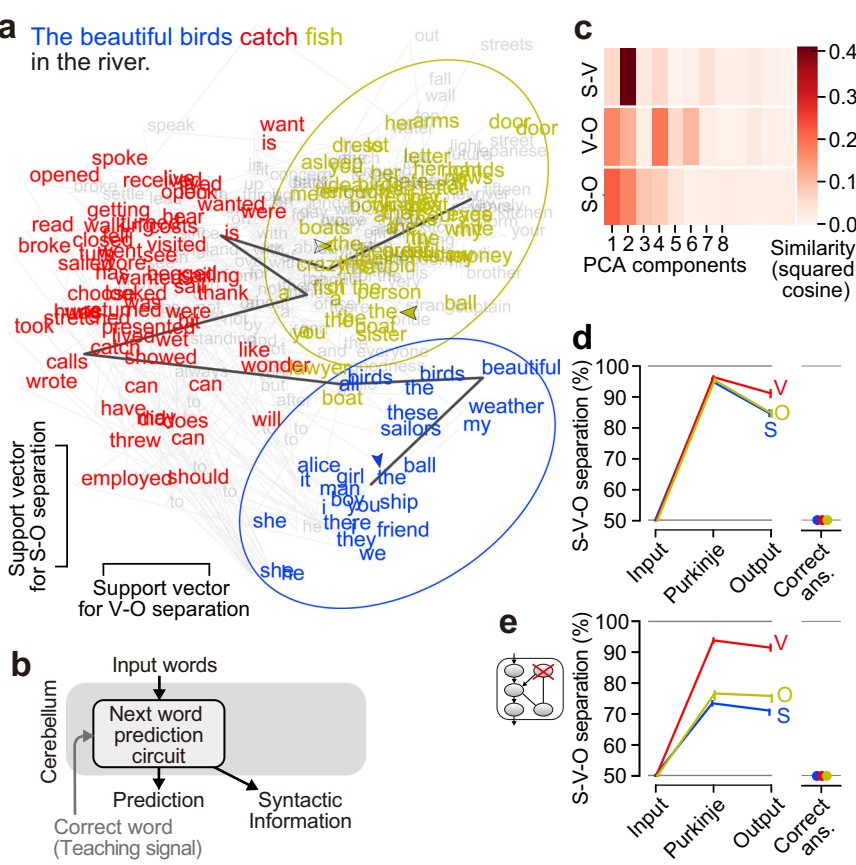

**Fig. 3 | Syntactic processing by the cANN. a** Visualization of neural activity of Purkinje cells after inputting a subject (blue), verb (red), and object (yellow), in the dimensions used for classification. The activity transition for a sentence with a longer subject (three words) is highlighted (dark gray trace). **b** Inputs/outputs for syntactic processing. **c** Similarity analysis of the dimensions of S-V, V-O, and S-O

separations with PCA dimensions (median). **d** Syntactic information contained in the feedforward-input, Purkinje, feedforward-output cells, and correct answer signal (n = 20 cANNs; median and IQR). The chance level of separation is 50% (gray horizontal line). **e** Syntactic information when the recurrent signal was blocked (n = 20 cANNs; median and IQR).

the recurrent pathway was designed according to the cerebellar anatomy, where connections from Purkinje cells to output cells are convergent, and connections from output cells to input cells are divergent. We set the number of output cells to 48 and the number of input cells to 192 in the recurrent pathway, so that Purkinje cell signals are compressed and decompressed through this pathway. Given that plasticity has been reported for various connections in the cerebellum[61], we assumed plasticity in the recurrent pathway. This cANN variant acquired the equivalent language functions to the original cANN (median of correct prediction rates, 36.5%; median of syntactic separation accuracy, S, 93.4%, V, 95.1%, O, 95.0%; $n = 4$). This demonstrates the robustness of cANN's language functions despite cell number variations in the recurrent pathway.

The second cANN variant, with inhibitory-restricted Purkinje-output connections, also achieved equivalent language functions (median of correct prediction rates, 36.8%; median of syntactic separation accuracy, S, 94.8%, V, 96.2%, O, 95.5%; $n = 8$). This demonstrates cANN's robustness regarding sign of connection between the Purkinje cells and the output cells, suggesting that such constraints do not hinder information processing. This aligns with the physiological observation that, although Purkinje cells have only GABAergic connections to the output cells, they can both inhibit and disinhibit the output cells by increasing and decreasing their firing rates (i.e., the sign of the downstream signal is not limited).

In the original cANN, the distribution of the synaptic weights of input-Purkinje connection resembled a normal distribution (Fig. 4a). If considering the greater-than-zero portion of the weights ($w > 0$; black in Fig. 4a) as the weights of parallel fiber-Purkinje cell synapses (i.e., direct excitatory input cell-Purkinje cell connection), this distribution closely aligns with the physiological distribution of parallel fiber-Purkinje cell synaptic weights, which can be approximated by a half of a normal distribution[62,63]. However, a sharp peak with zero synaptic weight ($w = 0$; silent synapses) was observed in the physiological data but not in the cANN. Therefore, we created the third cANN variant combining excitatory-limited input-Purkinje projections (imitating direct projection) and inhibitory-limited input-Purkinje projections (imitating indirect projection via molecular layer interneurons). This cANN variant also achieved comparable language functions (median of correct prediction rates, 37.0%; median of syntactic separation accuracy, S, 94.5%, V, 96.6%, O, 95.5%; $n = 4$). Interestingly, the distribution of the excitatory-limited synapses exhibited the peak of silent synapses, closer to the physiological observation (Fig. 4b). These findings demonstrate the robustness of the cANN functions to stringent biological constraints, while highlighting the need for such constraints to replicate circuit details such as synaptic weight distributions.

## Convergent cANN with the biological Purkinje-output convergence

We examined the performance of another cerebellar cANN model with an anatomical constraint of convergence of the Purkinje-output connection[40,64] and a physiological constraint of population coding

of the output cells[57]. To this end, we modified the original cANN circuit (Fig. 1b) to have only 16 output cells in the feedforward pathway so that the activity pattern of the 16 output cells represents a single word by population coding. Then, the modified circuit predicts only a single word at a time, unlike the original cANN which could probabilistically predict multiple candidate words. To enable the output of multiple word candidates, based on the physiological observation that the cerebellum has a modular structure which allows for the simultaneous generation of multiple outputs, we treated this modified circuit as one module and created a new cANN model with 10 modules in a row (Fig. 5a). Since parallel fibers, projections from the input cells to Purkinje cells, are known to pass across many modules, we assumed the input cells are shared between the modules. This new cANN is referred to as a "convergent cANN", while the original cANN is referred to as a "non-convergent cANN".

For learning, based on the cerebellar theory that enables multiple modules to cooperate and achieve a complex function (MOSAIC model)[32,54,64,65], after each prediction, the prediction error signal was given solely to the module with the closest prediction to the correct answer (=the actual next word). Based on the cerebellar theory that the output cells and the correct answer signal should have the same dimensionality[29,53-55], the correct answer was set to be represented in 16 dimensions. To represent the correct answer of 3000 words in 16 dimensions, we assumed a compressed word representation in which similar words (e.g., ship and boat) are represented by similar signal patterns, as observed in the neocortex[66], which is the source of the correct answer signal[10,26,29]. To facilitate comparison with the non-convergent cANN model, the following analysis focused on the top five modules in terms of prediction accuracy at the final training stage (see Methods). Training the convergent cANNs with the training sentences (e.g., from classic novels), once again, significantly improved next-word prediction (Fig. 5b, c). The correct prediction rate using the top 5 modules was 26.4% (median; interquartile range (IQR), 25.7%–27.1%, $n = 10$; Fig. 5c, blue; 33.6% by 10 modules), indicating that the convergent cANNs can also acquire next-word prediction. Because the learning of the modules occurs independently, one after another, each convergent cANN module had only 1/10 of the learning opportunities compared to the non-convergent cANN, which may have resulted in lower prediction accuracy. In the example sentences (Fig. 5d, top), three different instances of "the" generated different activity patterns of Purkinje cell signals (Fig. 5d, the[1], the[2], the[3]), which led to different proper predictions. Specifically, after "the[1]" for "man", a representative convergent cANN predicted "general", "prince", "roll", "same", and "kind"; after "open the[2]" for "gate", it predicted "post" and "door"; and after "read the[3]" for "book", it successfully predicted "book" and "press". In total, the correct prediction rate of the noun after a verb was 16.0% (median; IQR, 14.9–17.0%), which was slightly lower than that of the non-convergent cANNs. In contrast, the correct prediction rate of prepositions after verbs was 65.0% (median; IQR, 60.8–66.7%), slightly higher than that of the non-convergent cANNs. These results indicate that the convergent cANN has the capability to make different proper

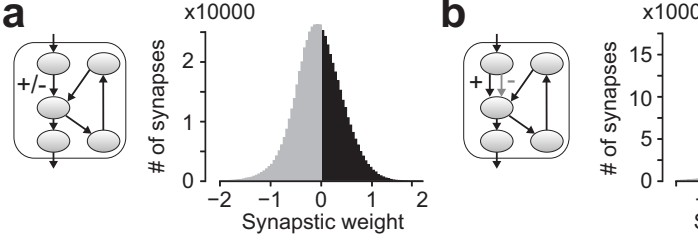

**Fig. 4 | Distribution of synaptic weights at the input cell-Purkinje cell connection. a** The original cANN's synaptic weight distribution without sign restriction at input-Purkinje synapses (left circuit diagram). The negative-weight component is in gray, and the positive-weight component is in black. **b** The synaptic weight distributions of a cANN variant with input-Purkinje synaptic signs restricted to either excitatory (black) or inhibitory (gray).

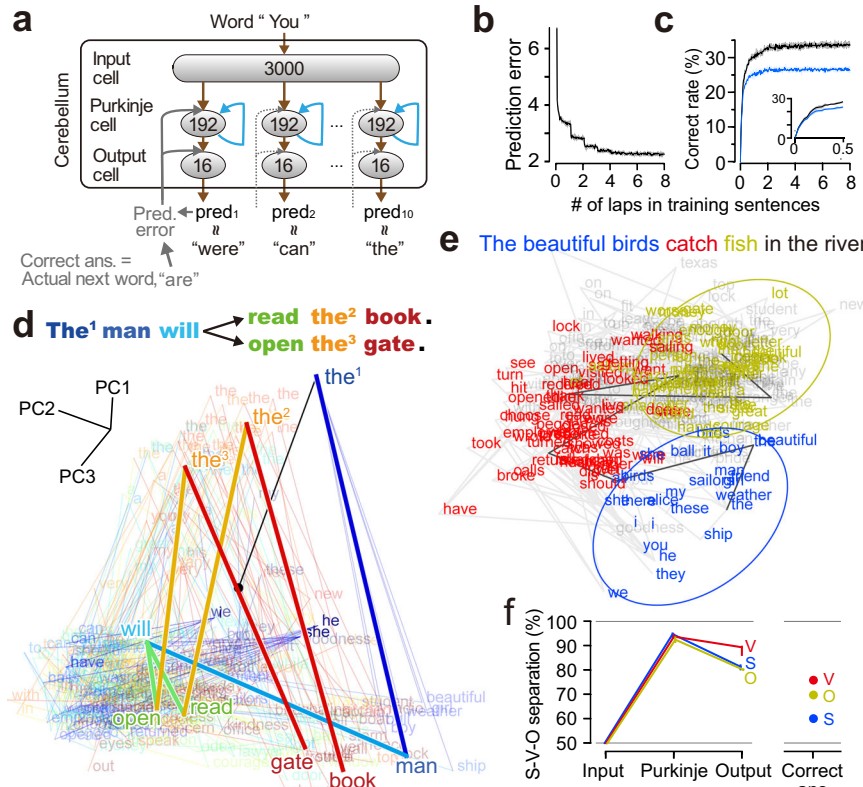

**Fig. 5 | Performance of the convergent cANN. a** Circuit design of the convergent cANN. The convergent cANN consists of 10 modules, each containing the same circuit as the original cANN in Fig. 1b, except for having 16 output cells. After each prediction, the prediction error is given solely to the module that produced the closest output to the correct answer signal of the actual next word (gray solid arrows). **b**, **c** Learning curves of 10 convergent cANNs for the prediction error (**b**; mean squared error) and percentage of correct predictions (**c**) for all the 10 modules (black) and for the top 5 modules (blue). **d** Visualization of the neural activity of Purkinje cells of a representative convergent cANN after input of each word, by PCA. Words are color-coded in rainbow from the beginning to the end of

the sentence, and spontaneous activity is indicated by the black dot. **e** Syntactic processing by the convergent cANN. Neuronal activity of Purkinje cells in the convergent cANN after input of a subject (blue), verb (red), and object (yellow) is visualized by the dimensions of the V-O and S-O classifications of the support vector machine. The solid black trace indicates the activity dynamics for the sentence, "The beautiful birds catch fish in the river". **f** Syntactic information contained in the feedforward-input, Purkinje, feedforward-output cells, and the correct answer signal ($n = 10$; median and IQR). The chance level of separation is 50% (gray horizontal line).

predictions according to context, although the prediction accuracy is slightly lower depending on the word type than the original non-convergent cANN.

We then examined the capability of syntactic processing in the Purkinje cells of the convergent cANN and found that signals for subject, verb, and object words were highly clustered and well separated (Fig. 5e). Comparing syntactic information along the feedforward pathway (Fig. 5f), we found that syntactic information was not above the chance level in the input layer, peaked at the Purkinje layer (S, 94.9%, V, 93.8%, O, 92.6%; almost equivalent to the non-convergent cANN), and was significantly degraded in the output layer ($p < 0.01$ for any of S, V, and O, Wilcoxon signed-rank test). Although the correct answer signal contained a small amount of syntactic information (Fig. 5f, right), the Purkinje cells in the convergent-cANN acquired a higher level of syntactic information, indicating that the convergent cANN has the capability to extract syntactic information beyond all external inputs. The important conclusion here is that our cerebellar models can acquire two cerebellar language functions, regardless of the dimensionality or the coding format of the output cells.

## Discussion

### Anatomical and physiological support for the cANN

Our cANN is highly consistent with a wealth of anatomical and physiological knowledge. This new model effectively accounts for the cerebellar language functions by incorporating the recently identified

recurrent pathway into the feedforward circuit, which is the fundamental structure of the conventional cerebellar circuit models. Regarding the recurrent pathway, numerous reports have demonstrated the existence of the vital part of the pathway, the output cell-input cell projections (that contain the direct projection[39–43] and indirect projections through the pontine or red nuclei[35–37]), and its functional significance on the cerebellar prediction[38,41,45]. The recurrent pathway has also been implemented in cerebellar modeling[67]. Consistent with this growing evidence for the significance of the recurrent pathway, we found that the pathway is essential for predicting the next word in a sentence and processing syntactic information in the cANN. We assumed that the Purkinje cell signal which is the source signal of the recurrent pathway is persistent until the next word arrives, based on the physiological observation that the predictive signal in Purkinje cells persists from the cue event until the actual event targeted for prediction occurs[38,49,50]. This persistent predictive signal enables the recurrent signal to last until the next word, preparing for integration with the next word information of the feedforward pathway. The time interval between words in natural speech and reading is approximately 300 milliseconds (200 words per minute), which aligns well with the reported time scale of predictive signal persistence in the Purkinje cells (over 500 ms)[38,49–52].

Building on these factors, our cANN has advanced the traditional cerebellar circuit model in two key aspects. First, while the cerebellum has been modeled as a feedforward circuit[63,68–74], the cANN

incorporates the recurrent pathway (Supplementary Fig. 3). The recurrent pathway is essential for two language functions, and its removal, whether before or after training, leads to significant impairment of these functions (Supplementary Figs. 1 and 3e). Second, in contrast to traditional cerebellar models based on the two-layer perceptron model where Purkinje cells are viewed as the output layer (Supplementary Fig. 3), the cANN treats Purkinje cells as the intermediate layer[57]. This redesign allows the Purkinje cells to engage in processing similar to that observed in the intermediate layers of ANNs of modern AI, such as feature extraction and data compression. Specifically, when the output layer of the cANN learns to predict the next word, the extraction of syntactic information becomes achievable in the intermediate layer. Moreover, to utilize this cerebellar syntactic information in other brain regions, such as the neocortex, an output pathway for the information in the intermediate layer is required. Based on our current anatomical and physiological knowledge, we propose two candidate output pathways that can be tested in future circuit studies (Supplementary Fig. 4a).

With respect to learning, the cANN utilizes prediction error-based learning, a method frequently employed in ANN models of the brain. Although this type of learning is widely used, the cerebellum is a rare region where its application is physiologically plausible[75]. The cerebellum is clearly considered to learn based on prediction errors in both motor and cognitive processing[10,29,40,46,76]. The cANN incorporates the inferior olive-climbing fiber pathway, which calculates and conveys prediction errors to the cerebellum (Fig. 1b). In prediction error calculation, there is no need for a retention mechanism for the prediction signal until the next word arrives, since the persistent predictive signal of Purkinje cells ensures that the circuit output persists. Also, a series of studies indicate that the cerebellum can learn more flexibly than classical cerebellar learning rules[44,46,47,61,77–82]. In particular, based on experimental findings and proposals that synaptic weights of Purkinje cells and cerebellar nucleus neurons are updated in the direction to reduce prediction error, our cANN updated synaptic weights[10,56,57].

Furthermore, we created cANN variants (including the convergent cANN) with further biological constraints, demonstrating the robustness of the two cerebellar language functions of the cANN. Together, our cANN is multilaterally consistent with our biological and physiological knowledge about the cerebellum and is capable of reproducing high-level cognitive functions of the brain.

## Significance of a common computational basis for the two functions

Because the cerebellum has a uniform cytoarchitecture[4,15,40,44,83–85], much conventional research has been directed toward elucidating the common circuit computation underlying all cerebellar functions. Although many descriptive ideas have been proposed, none have yet successfully explained all the diverse functions of the cerebellum[15]. For example, the internal-model theory states that the cerebellum is a predictive simulator of the external world. This theory can explain next-word prediction by replacing the external world as the prediction target with sentences and viewing the cerebellum as a sentence simulator[6–8,10,15,26]. However, explaining syntactic processing with the predictive functions of the theory is difficult and requires other ideas such as sequence processing[11,13,33].

Recognizing the limited reach of descriptive ideas to the common circuit computation, Diedrichsen, Ivry, and colleagues emphasized the need for a more abstract concept, like the network dynamics of neurons, to explain cerebellar computation[15]. Responding to precisely this need, we here demonstrated that the network dynamics visualized by the cANN can capture the common circuit computation underlying two major language functions. Considering next-word prediction as a language case of cerebellar prediction function (i.e., internal-model function of the cerebellum) and syntactic processing as a language case of sequence-processing function, we can conclude that a single

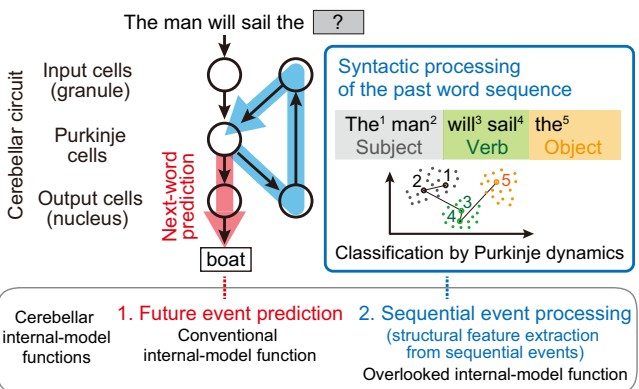

**Fig. 6 | Circuits for the two cerebellar language functions proposed by the cANN.** The entire circuit is trained to generate the output of next-word prediction (red). In parallel, to enhance the prediction accuracy, a syntactic processing circuit emerges upstream of the predictive output neurons (blue), so that Purkinje cell dynamics represent S-V-O syntactic information. Within the cerebellar internal model framework (bottom enclosure), next-word prediction corresponds to the future event prediction, aligning with the conventional understanding of the internal model function. Furthermore, syntactic processing corresponds to sequential event processing (structural feature extraction from sequential events), which is indicated by the cANN as another critical function of the internal model.

computational concept—the computation of our cANN—can unify both types of cerebellar general function, at least in language processing. The following general cerebellar computational mechanisms are indicated: (1) prediction-type (i.e., internal-model-type) circuits have inputs/outputs and error-based learning consistent with previous proposals (i.e., the circuit receives a series of events, produces predictions for one-step future events, and is trained by prediction errors)[6–8,10,25,26,29], (2) the recurrent signal is essential for long time-step dependent predictions, (3) sequence-processing-type circuits, depending on all the elements of the prediction-type circuits, spontaneously emerge upstream of the predictive output neurons. In summary, while traditionally the cerebellar internal model is considered to output predictive information about future events (Fig. 6, red), our cANN indicates that the internal model additionally processes sequences of events (i.e., extracting structural feature from past event sequences), representing another important, yet previously overlooked, output from the upstream stage of the internal model's predictive output (Fig. 6, blue).

Predictions by internal models are essential for various functions from motor control to cognition[4,5,10,15,29–32], and sequence processing is also of broad importance; for example, the motor sequences for tool use in humans have a hierarchical grammar-like structure (called "action grammar")[86,87]. Therefore, we propose that the circuit computation of our cANN is the generalized basis of the cerebellar computation that underlies a wide range of motor and cognitive functions.

## Differing roles of the neocortex and cerebellum in language function

The neocortex and cerebellum are tightly interconnected and cooperate to achieve sophisticated functions such as language processing[5,10,14,15,29,76,88–94]. However, it remains unclear how the functions of the two regions differ and how the roles are shared. When combined with clinical findings about language disorders, our cANN provides insights into the differing roles of the neocortex and cerebellum. Although the cerebellar role is limited in adults[15,95] and so the language functions of the cerebellum are considered subordinate to those of the neocortex, in children, cerebellar language disorders cause permanent and severe deficits[18–21], including agrammatism, suggesting that the development of the neocortex requires cerebellar support[15,18,19]. The ability of the cANN to extract syntactic information

independently from the neocortex suggests that syntactic information is sent from the cerebellum during development. If the neocortex can associate S-V-O information with the words themselves to produce information such as word class, this process gradually becomes self-sustaining, consistent with the clinical observations. Segmenting sentences according to S-V-O information may allow the neocortex to process sentences more efficiently[96] (hierarchical processing, Supplementary Fig. 4b, c). Considering that the cerebellum supports the development and maturation of other cognitive functions of the neocortex[10,15,97,98], the developmental importance of sequence processing by the cerebellum may generalize to other cognitive functions in addition to language processing.

## Human-characteristic cognition revealed by biologically constrained ANN models

ANN models can be categorized according to whether they comply with biological constraints, and if the functions they realize are human-characteristic cognitive functions, for which neuronal data are very limited (e.g., language processing, logical thinking, or social processing). Until now, no brain circuit model has been developed to fulfill both criteria: adhering to biological constraints and performing sophisticated human-characteristic cognitive functions. However, many models exist which fulfill one or the other demand.

For non-human-characteristic functions (including less complex cognitive functions), numerous ANN models have been proposed, with some adhering to biological constraints and others not. Biologically-constrained ANNs have been developed for regions like the cerebellum and basal ganglia, realizing various basic sensorimotor functions including decision-making[68–73,99–101]. ANNs that do not account for essential biological constraints, such as circuit connectivity between cell types, have been created for the cerebral neocortex using simple recurrent neural networks (e.g., three-layer recurrent neural network)[102–105] and for the visual cortex utilizing deep-learning circuits[106,107].

For higher-level cognitive processing such as language, far fewer ANNs have been proposed, and none of them are biologically constrained. Namely, models of neocortical language processing are based on language-processing AI circuits, which deviate from actual brain anatomy and physiology (e.g., transformer, LSTM)[1–3,108]. In contrast, the cANN is designed based on the cerebellar local circuit and successfully realizes language functions of the brain area. This is a pioneering proposal of a biologically-constrained brain model capable of human-characteristic cognitive functions.

## Future directions

We anticipate a prominent trend in neuroscience towards the development of biologically constrained ANNs to investigate human-characteristic cognitive functions, that are challenging to study in animal experiments. This study using cANN is a pioneer in such research.

Biologically constrained ANNs also have potential for clinical applications. The following insights gleaned from the findings of the cANN serve as an example. According to previous studies, when two functions share a circuit, training one function can improve the other, even in language functions[86,87]. Our findings predict that training in predicting the next word will improve the ability to process sentences with complex syntax, leading to the development of training in language processing and rehabilitation of language dysfunction.

Much remains to be elucidated about the grammatical processing performed by the cerebellum. Exploring the cANN's capacity beyond S-V-O syntactic processing, including its potential for word class recognition (suggested by Supplementary Fig. 2a) will offer valuable insight into cerebellar grammar processing.

Interestingly, the convergent cANN, which incorporates the further constraint of convergence in Purkinje-output connection into the original cANN, differs substantially from the typical circuit design of language-processing AI. Typical AI designs use compressed word representations containing semantic information for input and the sparse one-hot word representation for output. This input format aims to boost output accuracy by providing additional information at the input stage, in contrast to the output format, which is designed to facilitate listing multiple next-word candidates and accommodate sentences with various grammatically correct branches. Conversely, the convergent cANN employs sparse coding for input and a compressed word representation for output. This design can be interpreted as aiming to learn information-rich output from information-poor input, proposing a novel brain-inspired AI circuit. Future research is anticipated to uncover the potential of this circuit design.

# Methods
## Simulation environment

Code was written in Python and executed with Google Colab (Pro). The source code is available on GitHub (https://github.com/cANN-NLP/NLP_codes).

## Circuit design of cANN

The cANN was created with Google's deep learning libraries, TensorFlow and Keras (version 2.9 and 2.11). To replicate the local circuitry of the cerebellum, the cANN comprised three layers—an input layer (granule cells), a middle layer (Purkinje cells), and an output layer (cerebellar nuclei neurons)—with a recurrent pathway from the output layer to the input layer (Fig. 1b; Apps and Garwicz, 2005; Houck and Person, 2014; Ankri et al., 2015; Houck and Person, 2015; Gao et al., 2016; Raymond and Medina, 2018; Ohmae et al., 2021). In addition, the cANN implemented the climbing fiber pathway which is known to deliver prediction errors to the cerebellum in both motor and cognitive functions (gray, Fig. 1b) (Ito 2008, *Nat Rev*; Sokolov et al. 2017, *Trends Cogn*; Moberget et al. 2014, *JNS*). According to cerebellar learning theory that the cerebellum learns to align its output (prediction) very faithfully with the correct answer signal (Doya, 2000, *Curr Opin*; Ito 2008, *Nat Rev*), and that the representation of cerebellar output and correct answer signal have the same dimensions (Kawato 1999, *Curr Opin*; Kawato et al. 2011, *Curr Opin*), the output layer and the correct answer signal were set to have the same dimensionality. Then, the prediction error is calculated by subtracting the prediction from the correct answer signal.

The next-word prediction circuit was designed according to previous proposals, where words are fed sequentially into the circuit then, at each step, the circuit outputs a prediction of the next word, and information about the correct word information is given used as a teaching signal to improve future predictions. In addition to the circuit connectivity, this biological constraint on the inputs given to the circuit is also critical for simulating the acquisition of function, because, in general, the difficulty of learning is determined by the information supplied in the input signals, including error signals.

Integration of information between the feedforward and recurrent pathways occur as follows: When a sentence begins, the Purkinje cells receive information about the first word from the feedforward input cells and send their activity to the recurrent pathway (and to the feedforward output pathway). Because the Purkinje cell signal is persistent, the recurrent-input cell signal also persists until the feedforward-input cells receive the second word. The Purkinje cells then integrate both signals (i.e., the second word from the feedforward-input cells and the first-word information from the recurrent input cells) and transmit their new signal to the recurrent pathway. In this way, the Purkinje cells sequentially integrate a newly entered word with the previous words. We assumed that the feedforward and recurrent input cells are distinct cell populations because the cerebellum has an enormous number of granule cells in the input layer and individual cells receive only a small number of inputs (on average, only four).

To facilitate the investigation of information processing in this network, we assumed that word representation is coded sparsely in the input layer and the correct answer signal (each of 3000 words is represented by one of 3000 cells; e.g., one cell encodes 'I' and another cell codes 'the'). Then, the immediately preceding word, which is the input to the circuit, and the correct next word (for prediction error) are represented by an activity state in which only one cell is 1 and the rest are 0 (known as a one-hot representation). Theoretically, this representation is known to contain no information associated with the word, such as the syntax or meaning of the word (confirmed in Fig. 3d; cosine similarity is 0 for all word pairs); this allowed us to study the learning and information processing of cANNs in an environment where there is no information supplied externally other than word information.

The 3000 words represented by the 3000 cells were the 3000 most-frequent words in the training sentences. All the other words that appeared less frequently were grouped together as "unknown words" and assigned to the 3001st cell. For the prediction of the next word, the word encoded by the most active output cell denoted the first prediction candidate. Note that in the analysis, even if a prediction of "unknown" was correct, it was not considered a correct prediction of the next specific word so was excluded from the calculation of the correct prediction rate. The correct prediction rate is the percentage of matches between the top five prediction candidates and the correct next word (i.e., this selection of the five candidates was done for analysis purpose only).

Circuit connections denoted by arrows in Fig. 1b implement $F(\mathbf{W}x + \mathbf{b})$. For connections from N presynaptic cells to M postsynaptic cells, x is an (N x 1) vector representing the activity of the presynaptic neurons, $\mathbf{W}$ is an (M x N) matrix representing the synaptic weights from the presynaptic neurons to the postsynaptic neurons, $\mathbf{b}$ is an (M x 1) vector representing the spontaneous activity of the postsynaptic neurons, and F is the function that converts the neuron's input to its output (firing frequency) using the leaky Rectified Linear Unit (leaky ReLU; $y = x$, if $x >= 0$, and $y = \alpha * x$, if $x < 0$). For F from the input cells to the Purkinje cells and the output cells to the input cells, $\alpha = 0.14$, and for F from the Purkinje cells to the output cells, $\alpha = 1.0$. If alpha is 0, the input/output of the neuron is highly nonlinear; if alpha is 1, the input/output is perfectly linear ($y = x$). Although alpha could be modified quite flexibly to train the circuit, the cerebellum was assumed to be quite linear with Purkinje cells and output cells having high spontaneous firing rates (i.e., linear computing; Raymond and Medina, 2018). Since no functionally significant learning has been reported at this time for synaptic weights from the Purkinje cells to the recurrent output cells and from the recurrent output cells to the recurrent input cells, synaptic weights were assumed to be for relaying information (i.e., W = the identity matrix).

The prediction output was compared with the desired output of the true next word (i.e., the ground truth) to calculate the prediction error (E), using the following formula:

$$\text{Prediction error} = \text{cross entropy}(\mathbf{T}, \text{softmax}(\mathbf{O})) \qquad (1)$$

where $\mathbf{T}$ is the ground truth of the next word in the one-hot representation and $\mathbf{O}$ is the activity of the output cells. The softmax function normalizes the outputs so that the sum of the outputs is one and converts them to probabilities ($\mathbf{P}$), and the cross entropy is the information of the difference between $\mathbf{T}$ and $\mathbf{P}$:

$$\mathbf{P}_i = \text{softmax}(\mathbf{O}_i) = \exp(\mathbf{O}_i)/\sum \exp(\mathbf{O}_i), \text{ for } i = 1, \ldots, 3001 \text{ (index of cells)} \qquad (2)$$

$$\mathbf{E} = \text{cross entropy}(\mathbf{T}, \mathbf{P}) = \sum \mathbf{T}_i \log(\mathbf{P}_i) = \log\left(\mathbf{P}_j\right) \qquad (3)$$

for $\mathbf{T}_i = 1$ (i = j) and $\mathbf{T}_i = 0$ (i ≠ j), where j is the index corresponding to the next word. The prediction error was used to update synaptic weights on the input (i.e. dendritic) side of the Purkinje cells and the output cells (Fig. 1b, gray; Aizenman et al. 1998; Sokolov et al. 2017). Encouraged by recent discoveries and proposals suggesting stochastic gradient descent of synaptic updates (i.e., in the direction to reduce prediction error) in the cerebellum (Bouvier et al. 2018; Shadmehr 2020), we assumed a gradient-descending update:

$$\mathbf{W}_{ij} = \mathbf{W}_{ij} - \varepsilon \frac{\partial E}{\partial \mathbf{W}_{ij}} \qquad (4)$$

where W represents the synaptic weights and ε is the learning rate.

To make the recurrent pathway more powerful, the number of recurrent input cells could be increased, but increasing the number to more than 192 did not improve the accuracy of word prediction or classification of the syntactic information. This suggests that more sentences are needed to train a greater number of synapses.

## Circuit design of convergent cANN

Following the anatomical convergence of Purkinje-output connection, we designed the convergent cANN, in which we set the number of the feed-forward output cells to 16. Based on the cerebellar learning theory mentioned earlier that the representation of cerebellar output and correct answer have the same dimensions, we assumed that the cANN received 16-dimensional correct answer signal. To represent the correct answer of 3000 words in 16 dimensions, we employed a compressed word representation, akin to that in the neocortex (Huth et al. 2016, *Nature*). Concretely, we adopted the 100-dimension word representation, made by Stanford University group's GloVe algorithm and Wikipedia text dataset (glove-wiki-gigaword-100), and reduced the 100 dimensions to 16 using PCA. Consequently, each word was represented as a unique 16-cell activity pattern (a point in 16-dimensional space, fixed throughout the training). The candidate predicted by the 16 output cells was defined as the word whose activity pattern is closest to that of the output cells (≈, in Fig. 5a). Since the 16 output cells of one module can output only one word as a prediction, 10 modules were arranged in parallel with a shared input cell layer to replicate the modular structure of the cerebellum. For learning, based on the cerebellar learning theory of how multiple modules cooperate to achieve a complex function (Jacobs et al. 1991; Wolpert & Kawato, 1998; Wolpert et al. 1998, *Trends Cogn Sci*; Haruno et al., 2001; Kawato, Ohmae, et al. 2021), we set only the module with the closest prediction to the correct answer receives the prediction error signal (gray solid line, Fig. 5a). Although only one module learns after each prediction, with training on a large number of predictions, all modules should have the chance to make the best prediction and receive the prediction error (due to this, each convergent cANN module had only 1/10 of learning opportunities compared with the original non-convergent cANN, which may had resulted in slightly lower performance of the convergent cANN).

In the analysis, we examined the five candidate words of the convergent cANN to compare with the original cANN. To obtain the five candidates, we opted for training the 10-module cANN and selecting the five best-performing modules, as this design resulted in higher prediction accuracy than training the 5-module cANN and using all five modules. This difference in performance is attributed to the numerous grammatically correct branches in sentences (e.g., various adverbs and verbs can follow a subject) that five modules were inadequate to encode. The top five modules were selected based on prediction accuracy during training (in the last 1/8 lap). The differences in prediction accuracy between modules were highly consistent between training data and validation data.

## Circuit training

Twenty cANNs were trained independently. To train the cANNs, we used 88711 sentences from the Gutenberg Corpus (novels such as *Moby-Dick, Alice in Wonderland,* etc.) and the Brown Corpus of the Natural Language Toolkit (NLTK, version 3.8). Sentences that were too short (fewer than three words) or contained too many unknowns (four or more) were excluded. Next-word prediction was terminated at the 16th word if the sentence was too long. 90% (79444) of the sentences were used for training and the remaining 10% were used to evaluate the prediction performance during training. Training was terminated when the prediction performance plateaued (at the end of the eighth lap of the 79444 sentences).

## Data analysis

For the 20 cANNs, the median was used as the representative statistical feature and the interquartile range (25th and 75th percentiles) was used for the error bars, in all cases except where noted. To analyze the performance of the cANNs at next-word prediction and syntactic processing after training, we used relatively simple sentences from online sites (https://sentencedict.com/; https://lengusa.com/; https://www.rong-chang.com/nse/) because the novels in the NLTK corpus contain many sentences that are not syntactically common (e.g., conversational calls). We selected 190 sentences containing verbs that the cANNs could handle (i.e., verbs included in the 3000 encoded words). We added the sentence "The man will sail the boat," which was used by Lesage and colleagues to examine language processing in the cerebellum (Lesage et al. 2012), and the similar sentences "The man will read the book" and "The man will open the gate." All test sentences are available in a text file, named as test_sentences.txt, on GitHub (https://github.com/cANN-NLP/NLP_codes). In the training sentences, "read" and "open" appeared more frequently than "sail" (82 occurrences of "sail", 214 of "read", and 317 of "open").

The correct rate for next-word prediction was defined as the probability that one of the first to fifth candidate words would match the correct next word (ground truth). We selected up to the fifth candidate because that is the standard for the Google deep learning libraries used in this study (keras.metrics.TopKCategoricalAccuracy) and is also the standard in recent AI language processing. To examine the contribution of the recurrent signal, the recurrent signal was blocked by setting the synaptic weight from the recurrent-input cells to Purkinje cells to 0.

To analyze the dynamics of activity of the 192 Purkinje cells, 1328 words in the 190 sentences were fed sequentially into the cANN, and we measured the corresponding 1328 activity states in 192 dimensions of the Purkinje cells. To visualize the transition of Purkinje cell activity in a sentence, the 192 dimensions were compressed into two or three dimensions that maximally preserved the variance of the 1328 points, using principal component analysis (PCA; Python's online source-code library, scikit-learn 1.2). To evaluate the accuracy of classification of S-V-O syntactic information in the neuronal activity, we measured the percentage separation of S vs. V/O, V vs. S/O, and O vs. S/V with a nonlinear support vector machine (radial basis function kernel, scikit-learn 1.2). For this, 90% of the 190 evaluation sentences were used to determine the support vector and the remaining 10% were used to obtain the classification accuracy (%); this process was repeated 256 times to obtain the average accuracy (%). To visualize S-V-O syntactic information in Fig. 3a, we used the dimensions of the linear support vector machine, which are best suited for S-V, V-O, and O-S separations (coordinates of vectors orthogonal to the hyperplane best suited for the separations, obtained by svm.coef_ of scikit-learn). In the comparison of the optimal dimensions of the linear support vector machine with the PCA dimensions (Fig. 3c), the similarity index was calculated as the squared dot product of the unit vectors representing the dimensions (squared was used because the sum of the 192 dimensions is one).

## Reporting summary

Further information on research design is available in the Nature Portfolio Reporting Summary linked to this article.

## Data availability

No experimental data was generated in this study. The simulation data generated in this study have been deposited in the Open Science Framework database (https://osf.io/bwpnk/?view_only=7a9ae0981c2e4f8a921161dda9d079bd). The values of the data points displayed in this study are provided in the Supplementary Information/Source Data file. Source data are provided in this paper.

## Code availability

Codes were written in Python and executed with Google Colab (Pro). The source codes are available on GitHub (https://github.com/cANN-NLP/NLP_codes; https://doi.org/10.5281/zenodo.10257296)[109].

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

## Acknowledgements

We thank Dr. Javier Medina for multiple rounds of brilliant advice and Dr. Mitsuo Kawato, Dr. Tatsuo Sato, Dr. Tatiana Schnur, Dr. Nuo Li, and Dr. Shigeru Kitazawa (listed in the chronological order that the comments were provided) for invaluable insights and comments that improved the manuscript. We also thank Mrs. Zoha Hassan for editing to select subtle nuances of language, Dr. Masashi Kimura (via MENTA: https://menta.work/) for continuous helpful advice on programming, and Dr. Lindsay Bremner (via NeuroEdit: https://www.neuroedit.com/) for excellent editing. This work was supported by grant to SO from the National Institutes of Health (R34 NS118445).

## Author contributions

Conceptualization, S.O.; Data curation, S.O.; Formal analysis, S.O. and K.O.; Funding acquisition, S.O.; Investigation, S.O.; Methodology, S.O.; Project administration, S.O.; Resources, S.O.; Software, S.O.; Supervision, S.O.; Validation, S.O.; Visualization, S.O. and K.O.; Writing—original draft, S.O. and K.O.; Writing—review & editing, S.O. and K.O.

## Competing interests

The authors declare no competing interests.
