## [Peer Review File · Nature Communications]

Emergence of syntax and word prediction in an artificial neural circuit of the cerebellumReviewers' comments:

Reviewer #1 (Remarks to the Author):

General Comments

This manuscript proposes a neural network model with some features similar to the architecture of the mammalian cerebellum (so called cAAN). The model is used to explore aspects of language function, with the finding that training the model results in a modest ability to predict the next word and a second function, syntactic recognition. The latter is potentially the more intriguing aspect of the results, as the cAAN was trained on word prediction but not syntactic concepts. Therefore, what the authors refer to as a “biologically” constrained network shows some language processing properties. The manuscript is interesting, exploring the potential for language processing in AANs and their relation to neural circuits. The main concerns are two: 1) does the network emulate anything sufficiently like or specific to the cerebellum and 2) does the cANN provide actual insights into the cerebellum’s role in language processing.

The first concern is that the three-layer network with a recurrent pathway is not specific to the cerebellum. There are numerous structures in the brain that would easily fit that architecture. In fact, it is hard to think of a major circuit in the brain that can’t fit that model. Furthermore, the basic architecture of the three-layer network utilized seems a particularly poor match for the cerebellum. While the input to Purkinje cell layer shows a convergence that is like the cerebellar cortex, the Purkinje layer to nuclear neurons is modeled as a large divergence. This is not anatomically correct for the corticonuclear pathway and why this was chosen is unclear. Is the divergence important for the network to function? In fact, that convergence of Purkinje cells onto nuclear neurons is a major problem in cerebellar physiology and the cANN misses this aspect completely.

A second architecture issues is the recurrent pathway, which also seems to have limited relationship to the actual nucleocortical architecture. Why is the circuit so restricted to a subset, why 192 cells, and why are the synaptic weights fixed? Further, is there any evidence that this nucleocortical pathway plays such an important role in prediction, let alone of language. To place such an important role on one of the least studied and least understood aspects of the cerebellar circuit is a problem, when major features are left out such as the climbing fiber system or the massive inhibitory connections in the molecular layer. In this reviews opinion it would have been better to have tried to model the canonical cerebellar circuit.

The additional general concerns are the assumptions built into the “physiology” of the cAAN. Having the output cells assigned to a single word is non-physiological, as cell firing not be word specific or if the cerebellum would have any type of language representation at this level of specificity. That the recurrent circuit would hold a “word” in a type of buffer until the next word is imputed seems highly unlikely, at a single cell or population level. Also, using the best five word matches in the output is challenging. This would require another circuit to decide on the actual correct choice, “a deus ex machina”. Therefore, these types of processes seem divorced from what we know about cerebellar physiology.

Overall, AAN are powerful, and we know that they can be trained to do remarkable stuff. While the approach has potential value, the ANN would have to be much more specific to known cerebellar structure and physiology to help us understand how the cerebellum functions. Or the reverse would be exciting, that an ANN that had language function developed cerebellar-like properties when trained.

Specific Comments

1) The diagram in Figure 1b of the recurrent pathway is confusing, why are there two sets of 192 recurrent neurons? Shouldn't there be only one set? Also, as indicated above the nucleocortical pathway is not extensive, but the diagram suggests a perfect match with the 192 Purkinje cells, also unlikely. Need convincing that the recurrent architecture is viable.

2) Would allowing the weights of the recurrent pathway to change, help or hurt the model? Virtually all synapses in the brain are plastic, so this static property does seem realistic.

3) It was never clear if the architecture had the appropriate synaptic signs, that is the granule cell to Purkinje cell layer excitatory and is the Purkinje cell later to the nuclear neurons inhibitory? Do those features have any important consequences for network performance? Potentially more interesting, do they emerge without specifying these types of critical details?

4) Did the authors open up the trained network and find any interesting properties to how the synaptic weights were organized that are cerebellar-like?

Reviewer #2 (Remarks to the Author):

This paper provides an integrative perspective on two proposed cognitive functions of the cerebellum – lexical prediction and syntactic processing. It does so by demonstrating that an artificial neural network

with an architecture inspired by knowledge of cerebellar anatomical organization can demonstrate word prediction learning and the emergence of word class (noun, verb) knowledge organization. As strengths, this is a well-written paper that takes a clear theoretical stance that is innovative and potentially of broad interest. These strengths are tempered by some major concerns, which are noted below.

Major concerns:

1. Citations and discussions of the literature could be improved.

a. At numerous points, the text makes claims that are not fully supported by the cited literature. For instance, the text states that language comprehension after word recognition requires right Crus I/II, but most of the cited references focus on general theories of motor or cognitive control and none clearly provide empirical evidence of a required functional role. As another example, the text states that the cerebellum acquires language functions early in development compared to the rest of the brain, but evidence clearly in support of this claim is not provided in any of the three cited papers.

b. It would also be appropriate to cite the work of Porrill, Dean, and Stone (2003), who considered recurrent cerebellar architecture as an alternative to feedforward models.

c. The findings speak only to the most rudimentary elements of syntax (statistical separation of representations for nouns versus verbs), and only for a narrow set of conditions. It would be more appropriate to make claims about the emergence of word classes, rather than syntax more generally.

2. It is not clear that the cANN architecture is grounded in known neuroanatomy. The recurrent pathway is described as originating from the Purkinje layer, but the cited neuroanatomical findings instead argue for recurrent pathways that originate from the output nuclei.

3. Relatedly, the discussion of the cANN architecture fails to consider the source of the prediction error that drives learning. Since this is the actual word, it would seem that some mechanism would need to be in place to hold the predicted word activity until the actual word has occurred, and so it would be helpful to have some discussion about the feasibility of this timing relative to the speed of utterance programming, production, and perception.

4. The presentation of the methods and results makes it difficult to understand how well the results generalize to a wide variety of sentence types. The set of test sentences should be provided and accuracy data should be given for predicted words in different syntactic roles. The examples and findings demonstrate a natural outcome of the recurrent structure, which is that prediction is highest when there is a two-word phrase followed by a word with high lexical associations to the presented words (e.g., "read the book", "open the gate", "birds catch fish"). But presumably prediction accuracy would

plummet with slightly more complex syntactic structures (e.g., “read the short book”, “open the red gate”, “birds quickly catch fish”) that do not benefit from common co-occurrence. It would be informative to understand how the dynamics of the cANN would be structured in these situations.

5. The future direction is weakly developed and supported. It should be removed or considered in greater depth.

Here, we have quoted the Reviewers' comments in blue, and given additional numbering in red.

Reviewer #1 (Remarks to the Author):

General Comments

This manuscript proposes a neural network model with some features similar to the architecture of the mammalian cerebellum (so called cAAN). The model is used to explore aspects of language function, with the finding that training the model results in a modest ability to predict the next word and a second function, syntactic recognition. The latter is potentially the more intriguing aspect of the results, as the cAAN was trained on word prediction but not syntactic concepts. Therefore, what the authors refer to as a “biologically” constrained network shows some language processing properties. The manuscript is interesting, exploring the potential for language processing in AANs and their relation to neural circuits. **The main concerns are two: 1) does the network emulate anything sufficiently like or specific to the cerebellum and 2) does the cANN provide actual insights into the cerebellum's role in language processing.**

(1.1) The first concern is that the three-layer network with a recurrent pathway is not specific to the cerebellum. **There are numerous structures in the brain that would easily fit that architecture.** In fact, it is hard to think of a major circuit in the brain that can't fit that model.

(1.2) Furthermore, the basic architecture of the three-layer network utilized seems a particularly poor match for the cerebellum. While the input to Purkinje cell layer shows a convergence that is like the cerebellar cortex, **the Purkinje layer to nuclear neurons is modeled as a large divergence.** This is not anatomically correct for the corticonuclear pathway and **why this was chosen** is unclear. **Is the divergence important for the network to function?** In fact, that convergence of Purkinje cells onto nuclear neurons is a major problem in cerebellar physiology and the cANN misses this aspect completely.

In response to Reviewer #1's comments (shown in black):

(1.1) We appreciate Reviewer #1's comment regarding the generality of the three-layer network with a recurrent pathway in modeling brain structures. In addition, brain models that utilize 3-layer networks with recurrent pathways (RNNs) typically employ error-based learning (i.e., learning based on the output error, including prediction error). Hence, we would like to discuss this aspect as well. While it is true that three-layer recurrent neural networks (RNNs) trained by error-based learning have been applied to various brain structures, very few brain regions are biologically suitable for modeling with them. The cerebellum is one of the rare exceptions.

Indeed, the neocortex has been modeled using three-layer RNNs with error-based learning (e.g., Mante et al. 2013, *Nature*; Chen et al. 2021, *Cell*), but the neocortex is fundamentally a six-layer structure with multiple cell types intricately interconnected (Toga et al. 2006, *Nat Rev*), making it difficult to establish biological correspondences with a three-layer RNN. Moreover, evidence supporting error-based learning in local circuits of the neocortex

remains scarce and ambiguous. For this reason, neocortical models that utilize such three-layer error based-learning RNNs are not considered biologically constrained, and models that claim to be biologically constrained do not use such RNNs (Pulvermuller et al. 2021, *Nat Rev*; Chen et al. 2022, *Sci Adv*). Likewise, for the basal ganglia, three-layer RNNs have not been used in models that account for biological constraints (e.g., direct and indirect pathways; refs, Co and Wang, 2006, *Nat Neurosci*; O'Reilly and Frank, 2006, *Neural Comput*).

In contrast, the cerebellum can be considered as a three-layer structure (input-intermediate (Purkinje)-output) (see review papers of Medina & Mauk 2000, *Nat Neurosci* and Shadmehr 2020, *JNP*). Moreover, physiological studies have demonstrated that the cerebellum receives prediction errors via the climbing fiber pathway, which are essential signals for learning (the gray pathway in Figure 1b represents the climbing fiber pathway of the cANN, which will be explained in detail in the answer to the comment 2.3). Since Marr and Albus first proposed a perceptron model for the cerebellum in the 1970s, the cerebellum has been consistently modeled using a three-layer ANN trained by prediction error, adhering to biological constraints. Additionally, recent studies have demonstrated the importance of the recurrent pathway in the cerebellar functions (which will be explained in detail in the answer to the comment 2.2), and Michael Mauk and colleagues have incorporated this pathway into a cerebellar model (Khilkevich et al. 2018, *Sci Adv*) which resulted in a three-layer RNN model. Thus, the cerebellum possesses a unique suitability for modeling with three-layer RNNs trained by error-based learning.

We are grateful for Reviewer #1's comments, which have helped us to better appreciate the distinctiveness of our cANN within the broader context of ANN modeling for various functions. We have revised the "Introduction" (L50-60 after revision) to outline the current state of ANN modeling for language processing. Additionally, we have included a new paragraph in the "Discussion" (L468-489 after revision) to offer an overview of the current state of ANN modeling, emphasizing the uniqueness of our study in presenting a biologically constrained ANN model for human-characteristic cognitive functions, specifically cerebellar language functions.

(1.2) We appreciate the comments from Reviewer #1 on the anatomical validity of the cerebellar artificial neural network (cANN) model, especially regarding the divergence from Purkinje cells to output cells. To address the concern, we have created a convergent cANN, where the cerebellar cortex to nucleus projection converges, and have confirmed that it can acquire both language functions demonstrated by the original cANN (hereafter referred to as the non-convergent cANN). Therefore, we conclude that the divergent projection of the non-convergent cANN is not crucial for the network's language capabilities. We agree with the reviewer's comment that non-convergent design alone is not sufficient to demonstrate the anatomical validity of cANN. Therefore, we have created a new Figure 5 regarding the convergent cANN and included it in the revised manuscript.

In the following, we first explain why we originally presented the non-convergent cANN and then describe in detail the language processing capabilities of the convergent cANN, which is more anatomically plausible.

The non-convergent cANN was designed to examine the syntactic processing capability of the cANN circuit in the absence of external syntactic information. In other words, we

selected this design to evaluate the pure processing capability of the network itself, as described in more detail below. In the original manuscript, a brief explanation of this design selections was provided in the "Methods" section of L405-414 (L654-663 after revision).

The non-convergent cANN has two external signals: an input signal to the input cell (red in the right figure, "You") and a correct answer signal of the next word (red in the right figure, "are"). To create a condition where no syntactic information is provided by the external inputs, we employed 3000-dimensional sparse coding for these external signals and verified the absence of syntactic information in both signals (the revised Figure 3d, displayed below). Based on the cerebellar theory that the dimensions of the output cells and the correct answer are very similar (Kawato 1999, *Curr Opin*; Doya, 2000, *Curr Opin*; Kawato et al. 2011, *Curr Opin*), we assumed that the number of output cells is equivalent to the dimensions of the correct answer signal, which is 3000. This resulted in a divergence of the projection from 192 Purkinje cells to the 3000 output cells. By employing this design, we successfully demonstrated that the circuit of cANN can acquire syntactic information independently without any external support.

In the convergent cANN with 192 Purkinje cells and the 16 output cells, according to the population coding theory of the cerebellar output cells (Shadmehr, 2020), we assumed that a word is encoded by the activity patterns of the 16 output cells. Based on the cerebellar theory on the output dimensions mentioned above, the correct answer signal also needs to code 3000 words in 16 dimensions. Therefore, we assumed that the correct answer signal is represented by a compressed word representation, as found in the neocortex, i.e., similar words are represented by similar signals (Huth et al. 2016, *Nature*. This assumption is biologically plausible, since anatomically the neocortex is the source of the correct answer signal). As a result, the correct answer signal contains the word class information (For example, verbs are coded with similar signals and can be distinguished from other word types; see the right side of the new Figure 5f, displayed right). Therefore, with the convergent cANN, it was not possible to definitively determine whether the syntactic processing acquired by Purkinje cells was derived from internal circuit capability or external information. We consider the

presentation of the non-convergent cANN in the original manuscript to be significant as it demonstrates the circuit's inherent information processing capabilities.

Next, we explain how well the more anatomically plausible convergent cANN can acquire both language functions demonstrated by the non-convergent cANN, referring to the new Figure 5 (displayed below). In the convergent cANN, we assumed that one word is encoded by the population coding of the 16 output cells, and so we modeled the modular structure of the cerebellum so that the modules can predict multiple candidate words (see design in Figure 5a; Specifically, ten modules were modeled and the top five modules with the highest prediction accuracy were focused on in analysis). After training, the convergent cANN was able to adequately predict the next word of different word types according to context (for detailed accuracy see L334-363 of the revised manuscript). While prediction accuracy for some word types was slightly lower than that of the non-convergent cANN, we attribute this to the lack of training data as the circuit has multiple modules to be trained. For subject-verb-object (SVO) syntactic processing, the convergent cANN achieved nearly the same separation accuracy as the non-convergent cANN (Figure 5e,f; e, SVO clusters of Purkinje cell signals; f, SVO separation accuracy by signals in the feedforward pathway). Overall, we found that the more anatomically plausible convergent cANN can acquire full language functions demonstrated by the non-convergent cANN. Therefore, we conclude that the cerebellar language functions of the cANN are robust, regardless of the convergence or divergence of the cortico-nuclear projection.

We express our gratitude to Reviewer #1 for emphasizing the importance of discussing the

biological plausibility of the cANN. We, too, place great importance on ensuring the biological plausibility of the cANN. To this end, we have included a new Figure 5 in the revised manuscript that summarizes the design and analysis results of the convergent cANN. In addition, we have provided a detailed description in the Results section (L288-363), demonstrating that the convergent cANN is capable of acquiring two cerebellar language functions with high reliability. We appreciate Reviewer #1 for highlighting the necessity of this essential discussion.

(2) A second architecture issues is **the recurrent pathway**, which also seems to have **limited relationship to the actual nucleocortical architecture**. (2.1) Why is the circuit so restricted to a subset, why 192 cells, and why are the synaptic weights fixed? (2.2) Further, **is there any evidence** that this nucleocortical pathway plays such an important role in prediction, let alone of language. (2.3) To place such an important role on one of the least studied and least understood aspects of the cerebellar circuit is a problem, when major features are left out such as **the climbing fiber system** or the **massive inhibitory connections in the molecular layer**. (2.4) In this reviews opinion it would have been better to **have tried to model the canonical cerebellar circuit**.

Regarding (2.1), Specific Comments 1 and 2 both address the same issue, so we will provide a comprehensive response there and offer a brief reply here. Given the limited knowledge about the learning rules and information processing of the nucleo-cortical pathway, we assumed that the recurrent pathway in cANN does not process information except for non-linear thresholding in the input cells, resulting in a simple information relay pathway. This design allowed us to test whether the cANN circuit is capable of processing language even without assuming information processing in the recurrent pathway. The results demonstrate that cANN can adequately perform the cerebellar language functions. As the next step of our research, we created cANN variants with different numbers of cells and learning capacities in the recurrent pathway and evaluated their performance. Please refer to our responses to Specific Comments 1 and 2.

Regarding (2.2), there is direct evidence supporting the importance of the nucleocortical pathway in cerebellar prediction. In recent years, experimental studies have revealed the crucial role of the recurrent pathway in the input cell signals (Giovannucci et al. 2017, *Nat Neurosci*), gain control of predictive movement (Gao et al. 2016, *Neuron*), learning speed of prediction (Xiao et al 2022, *Cell Rep*), and Purkinje cell computation underlying prediction of event timing (Ohmae et al. 2021, *BioRxiv*, under revision in *Neuron*). The latter three studies, in particular, employed the eyeblink conditioning task (a classical task to study prediction in the cerebellum) and demonstrated the essential role of the recurrent pathway in prediction.

Regarding (2.3), Our cANN does include the climbing fiber system. This system is theorized to calculate the difference between the prediction and the actual event at the inferior olive and transmit the prediction error signal to the cerebellum for both motor and cognitive functions (Ito 2008, *Nat Rev*; Sokolov et al. 2017, *Trends Cogn*; Moberget et al. 2014, *JNS*). Specifically, in language processing, it has been proposed that prediction errors are conveyed through the

climbing fiber pathway (Fig. 1a). In this study, we followed these previous proposals, and the gray pathway in Figure 1b represents the climbing fiber system to send prediction error to the cerebellum. However, this was not explicitly stated in the text, which may have led Reviewer #1 to think that the climbing fiber system was omitted in the cANN. To address this issue, the climbing fiber system was explicitly described in the text (L107-108 in the Results section and L397-402 in the Discussion section) and figure legend to ensure clear understanding by the reader. We are grateful to the reviewer for bringing this to our attention.

The reviewer's comment that the molecular layer interneurons were not taken into account is accurate. However, we believe the omission does not constitute an oversimplification, because our cANN model follows an established tradition of the cerebellar modeling. The direct connection from input (granule) cells to Purkinje cells is excitatory, while indirect connections via molecular layer interneurons are inhibitory. Since both pathways can produce similar results in Purkinje cell signaling (e.g., the firing reduction of Purkinje cells can be induced either by activation of inhibitory indirect projections or by weakening excitatory direct projections, i.e., long term depression) (Schonewille et al. 2011, *Neuron*; Boele et al. 2018, *Sci Adv*), it has been a common practice in cerebellar modeling to exclude the inhibitory indirect projection (i.e., the Molecular layer Interneuron). In fact, none of the seven representative classical cerebellar models mentioned in Extended Data Figure 3 incorporate molecular layer Interneurons. Our original cANN model follows this established tradition. In addition, we created a cANN variant with a combination of an excitatory-restricted input-Purkinje pathway and an inhibitory-restricted pathway which imitated the indirect input-Purkinje projection via the molecular layer interneurons. This cANN variant preserved the performance of the language functions. Taken together, we consider that the molecular layer neurons are not essential for modeling the cerebellar circuit to reproduce the language functions.

Regarding (2.4), Reviewer #1's suggestion to model the canonical cerebellar circuit without the recurrent pathway is valuable. Indeed, we trained cANN to model the canonical cerebellar circuit (without the recurrent pathway) and found that it displayed severely poorer performance in both language functions, as briefly mentioned in L216-217 (L387-391 after revision). In addition, in Extended Data Figure 3, we carefully explained that all the classical cerebellar models without the recurrent pathway cannot account for the two language functions of the cerebellum, particularly syntactic processing. We appreciate Reviewer #1's comment making us aware of the importance of this discussion. We added explanation of the differences between cANN and these canonical cerebellar models in the Discussion section (L384-395).

(3) The additional general concerns are the assumptions built into the "physiology" of the cANN. (3.1) Having the output cells assigned to a single word is non-physiological, as cell firing not be word specific or if the cerebellum would have any type of language representation at this level of specificity. (3.2) That the recurrent circuit would hold a "word" in a type of buffer until the next word is imputed seems highly unlikely, at a single cell or population level. (3.3) Also, using the best five word matches in the output is challenging. This would require another

circuit to decide on the actual correct choice, “a deus ex machina”. Therefore, these types of processes seem divorced from what we know about cerebellar physiology.

Regarding (3.1), Reviewer #1 correctly noted the lack of physiological evidence supporting the idea that each output cell in the cerebellum is solely responsible for predicting a single word. The convergent cANN which achieves word representation by population coding, namely by activity patterns of the 16 output cells, is capable of addressing this concern, because population coding is considered as a biologically plausible representation form of the output cells (Shadmehr, 2020, JNP). We confirmed that the convergent cANN can also acquire the cerebellar language functions (see our response to Comment (1.2)). The important conclusion drawn from this is that our cANN's acquisition of cerebellar language functions is robust, regardless of the output format, whether the word is represented by population coding or specifically-assigned coding. To address these important discussions, we have included additional descriptions in the Results (L289-300) and Methods (L712-725) sections of the revised manuscript.

Regarding (3.2), Reviewer #1 expressed concern that the recurrent circuit may not be capable of holding word information as a buffer until the next word arrives due to its simple tri-synaptic structure. This concern can be rephrased as when transient information from Purkinje cells enters the recurrent circuit without a buffer function, the information will decay and lost before the next word comes. This concern is entirely reasonable and stems from our misleading statement. In Figure 1b, we wrote "one-word delay" in the recurrent pathway, and this may have given the impression that the recurrent pathway has buffer function to retains information until the next word comes, which is not accurate. We apologize for this misleading statement.

To begin with the conclusion, based on a physiologically plausible assumption that the source signal of the recurrent pathway from Purkinje cells is persistent, the buffer function of the recurrent pathway is not necessary. In our cANN, the signal of the recurrent pathway is a branching signal of the circuit computation for the output of next-word prediction from Purkinje cells. Experimental studies using the well-studied eyeblink conditioning showed that Purkinje cells generate predictive signals persistent on timescales of hundreds of milliseconds (Johansson et al. 2014, *PNAS*; Halverson et al. 2015, *JNS*; Ohmae et al. 2021, *BioRxiv*). Therefore, in next-word prediction, the persistent predictive signal of Purkinje cells is a physiologically plausible assumption. If the predictive signal is persistent, since this signal is the source for the recurrent circuit, the recurrent signal is persistent until the next word, eliminating the need for a buffer function in the recurrent pathway. The average speaking or reading speed of approximately 200 words per minute (i.e., 3.3 words/sec, or 300 milliseconds/word) corresponds with the timescale for predictive signal persistence in Purkinje cells (over 500 ms).

To avoid misunderstandings about signal buffering in recurrent circuits, we have rephrased “one-word delay” as “one-word persistence”, and carefully revised the statements and discussion accordingly. We have included our biological assumption and its discussion in the Results section (L109-113) and in more detail in the Discussion section (L376-383).

Regarding (3.3), Reviewer #1's concern that a special circuit ("a deus ex machina") is required to select the best five word matches from all output cell signals is understandable but it stems from a misunderstanding of the cANN processing. The selection of the top five words as prediction candidates is solely for the purpose of demonstrating the cANN's performance to the reader and is not related to the cANN circuit's processing. We understand that our description of the top five word selection in L 103-104 (in the original manuscript) might have caused confusion, as we did not emphasize that it is an analysis for presentation purposes only. To prevent further misunderstanding, we have revised the text (L127-129, L670-672) to ensure clarity and avoid misleading statements. We deeply appreciate Reviewer#1's thoughtful comments in bringing this critical point to our attention.

Overall, AAN are powerful, and we know that they can be trained to do remarkable stuff. While the approach has potential value, the ANN would have to be much more specific to known cerebellar structure and physiology to help us understand how the cerebellum functions. Or the reverse would be exciting, that an ANN that had language function developed cerebellar-like properties when trained.

In our study, we employed a novel approach utilizing biologically constrained ANNs to investigate circuit computations of human-characteristic cognitive functions that are difficult to address through animal experiments. We believe that this approach will become increasingly important trend in the future and pleased that Reviewer #1 also recognizes its potential value. Additionally, we believe that our responses have addressed any concerns regarding the inadequacy of biological constraints. Reviewer #1's insightful comments have significantly improved and clarified our manuscript's value. We deeply appreciate your contribution.

Specific Comments

1) The diagram in Figure 1b of the recurrent pathway is confusing, why are there two sets of 192 recurrent neurons? Shouldn't there be only one set? Also, as indicated above the nucleocortical pathway is not extensive, but the diagram suggests a perfect match with the 192 Purkinje cells, also unlikely. Need convincing that the recurrent architecture is viable.

The recurrent pathway of cANN emulates the nucleo-cortical pathway of the cerebellum. The nucleo-cortical pathway projects from the output cells in the cerebellar nucleus to the input cells (granule cells) in the cerebellar cortex. Two sets of cell populations of the cANN correspond to these output and input cells. Since little is known about the learning rules and information processing of the recurrent pathway, we made the simplest assumption for the cANN's recurrent pathway: there is no information processing other than intracellular nonlinear transformation in the input cells. With this assumption that information is simply relayed, the numbers of recurrent output and input cells are set to 192, the same number as Purkinje cells. We initially designed the cANN in this manner to test whether the cANN circuit could process language despite the limited learning and information processing capabilities of the recurrent pathway. Through this approach, we successfully demonstrated the cANN's capabilities. For the next step, we created cANN variants with varying numbers of neurons in the recurrent pathway

where learning takes place. We provide detailed descriptions of these variants in our response to Specific Comment 2.

Furthermore, the idea that the cerebellar recurrent pathway is not abundant is being overturned. Anatomically, abundant projections have been reported from different cell types in the cerebellar nucleus (Houck and Person, 2014; Ankri et al., 2015, *Neuron*; Houck and Person, 2015; Gao et al., 2016, *Neuron*; Low et al. 2018, *CellRep*). In addition, a calcium imaging observation indicated that more than half of the input cells contain output information of the cerebellum (Giovannucci et al. 2017, *Nat Neurosci*). We have cited this supporting physiological evidence and discussed the importance of the recurrent pathway in the Discussion section (L370-373).

2) Would allowing the weights of the recurrent pathway to change, help or hurt the model? Virtually all synapses in the brain are plastic, so this static property does seem realistic.

Firstly, we allowed plastic changes in the recurrent pathway while maintaining the number of output cells at 192 and input cells at 192. As a result, the correct prediction rates for the next word were almost equivalent to those of the original cANN (median of correct prediction rates, 36.9%). Syntax recognition was also equivalent to the original cANN (median of syntactic separation accuracy, S, 95.4%, V, 96.5%, O, 96.3%). We deduce that the recurrent pathway could not function beyond relaying information even with updatable synaptic weights, as new information is not integrated in this pathway.

Secondly, we enabled synaptic weight updates in the recurrent pathway with 48 output cells and 192 input cells. This design is more biologically plausible since it converges from 192 Purkinje cells to the 48 output cells in the recurrent pathway. This modified cANN showed performance almost equivalent to the original cANN (median of correct prediction rates, 36.5%; median of syntactic separation accuracy, S, 93.4%, V, 95.1%, O, 95.0%). In the recurrent pathway, it was necessary to compress the input from 192 Purkinje cells to 48 dimensions and then restore it to 192 dimensions, and we consider this transformation could be accommodated by appropriately modifying the synaptic weights.

Finally, we allowed synaptic weight updates in the recurrent pathway with 48 output cells and 384 input cells. This modified cANN also showed performance that was almost equivalent to the original cANN, but the correct prediction rate for the next word was slightly lower (median of correct prediction rates, 35.2%; median of syntactic separation accuracy, S, 93.6%, V, 96.9%, O, 94.9%). In theory, having more input cells for decompressing the recurrent signal would be beneficial (if half of the input cells were not used, this third version is equivalent to the second version). However, doubling the input cells also doubled the number of synaptic weights to be learned in the recurrent pathway. This may have caused insufficient training data, leading to slightly lower performance.

In conclusion, if synaptic weights in the recurrent pathway can be updated through learning, it is possible to compress the recurrent signal at the output cell layer and restore it at the input cell layer without degradation. We deeply appreciate the valuable comments of Reviewer #1 that led to this interesting finding. We included these results as performance of cANN variants in the Results section (L245-255).

3) It was never clear if the architecture had the appreciate synaptic signs, that is the granule cell to Purkinje cell layer excitatory and is the Purkinje cell later to the nuclear neurons inhibitory? Do those features have any important consequences for network performance? Potentially more interesting, do they emerge without specifying these types of critical details?

In the original cANN, there were no particular restrictions on the sign of synaptic weights. To address the reviewer's question, we conducted additional simulations.

Firstly, we modeled the input-Purkinje cell connection as a combination of positive-restricted (direct connection via parallel fibers) and negative-restricted (indirect connection via molecular layer interneurons) connections. This modified cANN exhibited performance equivalent to the original cANN (median of correct prediction rates: 37.0%; median of syntactic separation accuracy: S, 94.5%; V, 96.6%; O, 95.5%). This can be interpreted that the original cANN, which was able to represent both positive and negative weights, resulting in equivalent performance to the modified cANN variant with a combination of positive-restricted and negative-restricted connections.

Next, we restricted the Purkinje cell-output cell connection to negative weights. Since the original cANN could be both positive and negative, it was reasonable to consider that this restriction might affect performance. However, this cANN variant with negatively-restricted weights exhibited equivalent performance to the original cANN (median of correct prediction rates: 36.8%; median of syntactic separation accuracy: S, 94.8%; V, 96.2%; O, 95.5%). The negatively-restricted distribution of synaptic weights after training is shown in the figure below.

This result suggests that restricting the sign of the Purkinje cell-output cell connection to negative values does not significantly limit information processing. This may be consistent with the physiological observation that the output cells can display various firing patterns because increase of Purkinje cell firing can suppress the output cell activity and its decrease can disinhibit the output cell.

We deeply appreciate the careful comments of the Reviewer #1, which again led to these insightful simulation results. We introduced these results as performance of cANN variants in the Results section (L256-279).

4) Did the authors open up the trained network and find any interesting properties to how the synaptic weights were organized that are cerebellar-like?

In response to Reviewer #1's specific comment 4, we conducted an analysis of the synaptic weights of Input-Purkinje synapses. As shown in Figure (a) below, the synaptic weights after learning exhibited a distribution resembling a normal distribution. By considering half of the weights that are equal to or greater than zero as the synaptic weights of parallel fiber-Purkinje cell connections, this result closely aligns with the physiological distribution of parallel fiber-Purkinje cell synaptic weights, which can be approximated by a half of a normal distribution (Isope and Barbour 2002, *JNS*; Brunel et al. 2004, *Neuron*). However, physiological observations have demonstrated the presence of numerous silent synapses with a sharp peak at zero synaptic weight, which was not observed in the cANN's distribution.

To make the cANN more biologically realistic, we established two types of connections for the input-Purkinje cell synapses: positively restricted connections (emulating direct connections via parallel fibers) and negatively restricted connections (emulating indirect connections via molecular layer interneurons). Intriguingly, as shown in Figure (b), the synaptic weight distribution of positively-restricted synapses (black distribution) displayed a sharp peak at 0 weight, akin to the silent synapses in the physiological observation.

This cANN variant retains its language processing capabilities (median of correct prediction rates: 37.0%; median of syntactic separation accuracy: S, 94.5%; V, 96.6%; O, 95.5%), while its synaptic weight distribution more closely resembles the physiological observation. Although the original cANN is effective in revealing crucial computations of cerebellar language processing, these findings suggest that the implementation of more stringent physiological constraints is essential for accurately modeling the circuit in detail.

We have incorporated these fascinating results into the manuscript in the Results section (L264-279) with a new figure 4. We would like to express our gratitude to Reviewer #1 for suggestions that have improved the biological realism of the original cANN.

Reviewer #2 (Remarks to the Author):

This paper provides an integrative perspective on two proposed cognitive functions of the cerebellum – lexical prediction and syntactic processing. It does so by demonstrating that an artificial neural network with an architecture inspired by knowledge of cerebellar anatomical organization can demonstrate word prediction learning and the emergence of word class (noun, verb) knowledge organization. **As strengths, this is a well-written paper that takes a**

clear theoretical stance that is innovative and potentially of broad interest. These strengths are tempered by some major concerns, which are noted below.

First of all, we would like to express our sincere appreciation for Reviewer #2's high recognition of our work in modeling cerebellar language functions using cANN. The recognition has greatly encouraged us for the direction of our research. In response to Reviewer #2's valuable appreciation, we have included a comprehensive description in the Discussion section to provide a clear explanation of the impact of our research. This will allow readers to readily comprehend the strengths of this study as per your evaluation. Thank you so much once again for your valuable comments.

Major concerns:

1. Citations and discussions of the literature could be improved.

a. At numerous points, the text makes claims that are not fully supported by the cited literature. For instance, the text states that **language comprehension after word recognition requires right Crus I/II**, but most of the cited references focus on general theories of motor or cognitive control **and none clearly provide empirical evidence** of a required functional role. As another example, the text states that **the cerebellum acquires language functions early in development compared to the rest of the brain**, but evidence clearly in support of this claim is not provided in any of the three cited papers.

We are grateful for the reviewer's feedback, which highlighted the need to enhance our citations. In response, we have meticulously reviewed all citations to ensure that our paper is supported by accurate and relevant sources.

In reference to the reviewer's first example, we acknowledge that most of the original citations in L45-47 of the introduction are about neocortico-cerebellar cooperation in cognitive functions, rather than language. In addition, some important references related to this cooperation in language were not included in the original citations. This discrepancy occurred as we failed to verify the citations when revising the description of cognitive function to language function in the text. To rectify this oversight, we have updated the citations in L45-47 (L61-62 after revision) based primarily on the accurately cited literature in the later part of the introduction (L74-80 after revision). Now, the citations support the involvement of the right Crus I/II in the two language functions, cooperating with the left neocortical language area.

As for the second example, we have opted to remove the relevant portion of L47-48 (in bold), "The cerebellum has a large capacity for plasticity **and acquires language functions relatively early in development compared with the rest of the brain**^{8, 20, 21}", (L63 after revision). This decision was made because the statement is redundant with the subsequent sentence and the cited references do not sufficiently support the claim. We are grateful for Reviewer #2's astute observation, which has allowed us to enhance the accuracy and clarity of our manuscript.

b. It would also be appropriate to cite the work of Porrill, Dean, and Stone (2003), who considered recurrent cerebellar architecture as an alternative to feedforward models. We are grateful for Reviewer #2’s suggestion to cite the work of Porrill, Dean, and Stone (2003). Their significant paper presents a model of the cerebellum receiving recurrent motor command inputs and its contribution to motor control. This paper focused on the recurrent connections between the neocortex and the cerebellum, rather than the recurrent connections within the cerebellar local circuits. Therefore, we have appropriately cited this work in the first sentence of the 3rd section of the Discussion (L449-451).

c. The findings speak only to the most rudimentary elements of syntax (statistical separation of representations for nouns versus verbs), and only for a narrow set of conditions. It would be more appropriate to make claims about the emergence of word classes, rather than syntax more generally.

We acknowledge that our findings primarily address the most basic elements of syntax, focusing on the statistical separation of representations in terms of subject-verb-object. However, we believe this differs from word class recognition. Word class recognition involves identifying the grammatical category to which a word belongs, such as nouns, verbs, adjectives, or other parts of speech. In contrast, S-V-O syntax recognition requires the identification of subject (S), verb (V), and object (O) elements, which is essential for understanding a sentence's structure. Even for the same word, the cANN distinguishes between instances when the word serves as a subject and when it functions as an object, indicating its capability in S-V-O recognition rather than word class recognition. For example, in the revised Figure 3a (below), the article "the" is clearly within the subject area when appearing as a subject (blue arrowhead) and within the object area when appearing as objects (yellow arrowheads). Therefore, we consider the cANN, capable of differentiating SVO, captures the cerebellar grammatical processing, particularly the extraction of syntactic information. We did not previously describe these aspects explicitly, which may have led to confusion for Reviewer #2. We have now clarified this in the revised manuscript at L211-214, and replaced the Figure 3a with Extended Data Figure 2b to display clear separation between S (blue circle) and O (yellow circle) clusters.

3. Relatedly, the discussion of the cANN architecture fails to consider the source of the prediction error that drives learning. Since this is the actual word, it would seem that some mechanism would need to be in place to hold the predicted word activity until the actual word has occurred, and so it would be helpful to have some discussion about the feasibility of this timing relative to the speed of utterance programming, production, and perception.

We have assumed that the predictive signal generated by Purkinje cells persists until the input of subsequent word. This assumption is grounded in physiological evidence demonstrating that when the cerebellum predicts events, Purkinje cells generate enduring predictive signals until the actual event (i.e., unconditional stimulus) occurs. Notably, studies employing the well-established eyeblink conditioning paradigm have revealed that Purkinje cells generate a predictive signal in response to the conditional stimulus (e.g., sound) that anticipates the unconditional stimulus (e.g., air puff to the eye), and this predictive signal lasts until the actual unconditional stimulus takes place (Johansson et al. 2014, *PNAS*; Halverson et al. 2015, *JNS*; Ohmae et al. 2021, *BioRxiv*). Consequently, we surmise that when the subsequent word is presented, the predictive signal is still present, allowing the cANN to calculate the prediction error by subtracting both signals, thus eliminating the need for any specialized mechanism to preserve the predictive signal. The average speaking or reading speed of approximately 200 words per minute (i.e., 3.3 words/sec, or 300 milliseconds/word) corresponds with the timescale for predictive signal persistence in Purkinje cells.

We have described our assumption and its rationale in the Results section (L109-114) and provided a more in-depth explanation in the Discussion (L376-383 and L402-405). We are grateful for Reviewer #2's insightful comment, which has significantly improved our manuscript.

4. The presentation of the methods and results makes it difficult to understand how well the results generalize to a wide variety of sentence types. **The set of test sentences should be provided and accuracy data should be given for predicted words in different syntactic roles.** The examples and findings demonstrate a natural outcome of the recurrent structure, which is that prediction is highest when there is a two-word phrase followed by a word with high lexical associations to the presented words (e.g., “read the book”, “open the gate”, “birds catch fish”). But presumably prediction accuracy would plummet with slightly more complex syntactic structures (e.g., “read the short book”, “open the red gate”, “birds quickly catch fish”) that do not benefit from common co-occurrence. It would be informative to **understand how the dynamics of the cANN would be structured** in these situations.

In response to Reviewer 2's comment, we have created a new text file, named as test_sentences.txt, which contains the set of test sentences. This file is available on github (https://github.com/cANN-NLP/NLP_codes). This access information for this repository has also been included in the manuscript as well (L766-767).

In the original manuscript, we focused on the prediction of nouns (objects) after verbs, because this form of word prediction was thoroughly examined in Lesage et al (2012 & 2017), and the cerebellar involvement was established through functional imaging and TMS studies.

However, in response to the request, we have analyzed the accuracy of predicting other types of words. We found that the prediction accuracy for prepositions after verbs was 60.0% (median; 56.7-63.3, IRQ), for pronouns after verbs was 56.5% (52.2-60.9), and for nouns after prepositions was 43.0% (42.1-45.6). The prediction accuracy of adjectives after “be” verbs, such as “I was happy.” was 27.5% (25.0-30.0). These findings indicate that the cANN is able to predict the next word in various contexts. We have included this information in the Results section (L173-176).

To explore the dynamics of the cANN when dealing with sentences containing more complex syntactic structures, we visualized the dynamics of Purkinje cell activity for phrases such as for “read the short book” and “open the red gate” (below left; for a comparison with the original dynamics, Fig. 2c is placed right). Interestingly, we observed the differences between the² and the³ activities were retained in the “short” and “red”. However, as noted by Reviewer 2, the prediction accuracy significantly decreased in these cases. Specifically, after “read the short” for “book”, the cANN predicted “and, way, of, steps, time”(including “line” and “stories” in the top 10). Similarly, after “open the red” for “gate”, the cANN predicted “and, way, of, steps, time”. We presume that increased distance between the verb and the predicted word reduced the weight of the verb information, leading to the prediction of words more closely related to the preceding words, “short” or “red.”

To summarize, our revised analysis confirms the cANN's proficiency in predicting various word types, even though its accuracy decreases with more complex structures. Nonetheless, the cANN still maintains the dynamics related to cue information for prediction, indicating potential for further improvement to make proper predictions in complex sentence structure. We are grateful to Reviewer #2 for bringing this to our attention and for the careful review.

5. The future direction is weakly developed and supported. It should be removed or considered in greater depth.

We appreciate the constructive feedback we have received and have carefully incorporated it into our revised manuscript. To address the suggestions, we have elaborated on our descriptions and placed our cANN within the broader context of biological ANN research for human-characteristic cognitive functions. We also have highlighted the potential of our cANN for a brain-inspired AI circuit. Furthermore, we have included the discussion of the potential ability of the cANN for word-class recognition, which was raised by Reviewer #2. This valuable input enabled us to enrich our manuscript with additional insights and provide a more in-depth explanation of our research.

REVIEWER COMMENTS

Reviewer #1 (Remarks to the Author):

The authors have been very responsive to my many concerns, resulting in a much stronger and more convincing manuscript that the neural network architecture investigated has biological features that are “cerebellar-like” and can be constrained by cerebellar specific characteristics. To do this the authors have added simulation to address these concerns including on the recurrent pathway, using a model that better capture the convergence-divergence-convergence of the cerebellum, incorporating the synaptic sign, and incorporating inhibitory and excitatory inputs into the Purkinje cell layer. These additional models/simulations resulted in major changes to the manuscript and multiple new figures. Together these new results make the work more convincing and intriguing to cerebellar neurobiologists as well as to neuroscientists, more generally.

There are a few remaining questions. First, the prediction error into the network is equated with the climbing fiber system. Does this predictive error have any properties that would be somewhat like the rather unique climbing fibers and the complex spikes evoked in Purkinje cells? One thought is that the climbing fiber input is a type of non-linearity, and interesting they used a non-linearity to model the recurrent input.

Second, for the additional ways in which they modified the network in response to my concerns, the results were almost universally positive. While on one level this is good, it also begs the question. Is this a feature of these powerful neural networks, or is this a property of the cerebellar architecture?

Third, in lines 196-198 the statement is made that the cerebellum must acquire language processing without the cerebral cortex. This does not seem realistic and not certain way this claim is made. Whatever the cerebellum does, it does in concert with many other structures and certainly with the cerebral cortex for language processing. In the Discussion, the authors state that the cerebellum acquires language processing in concert with the cerebral cortex.

Fourth, line 382 comments that cerebellar predictive processing can be on the order of 500 sec. Here the work of Popa et al should be included that found evidence for predictive signals in Purkinje cell simple spike firing for long times including up to 2 seconds (Popa et al, 2012, J Neurosci 32(44):15345-58 and Popa et al, 2017 eNeuro 4).

Reviewer #2 (Remarks to the Author):

The authors have provided a very detailed response and made substantive changes that have greatly improved the manuscript, leaving no significant concerns.

The responses below address the reviewers' comments point-by-point (the reviewers' comments are quoted in blue for clarity).

Reviewer #1 (Remarks to the Author):

The authors have been very responsive to my many concerns, resulting in a much stronger and more convincing manuscript that the neural network architecture investigated has biological features that are “cerebellar-like” and can be constrained by cerebellar specific characteristics. To do this the authors have added simulation to address these concerns including on the recurrent pathway, using a model that better capture the convergence-divergence-convergence of the cerebellum, incorporating the synaptic sign, and incorporating inhibitory and excitatory inputs into the Purkinje cell layer. These additional models/simulations resulted in major changes to the manuscript and multiple new figures. Together these new results make the work more convincing and intriguing to cerebellar neurobiologists as well as to neuroscientists, more generally.

We thank Reviewer #1 for the thoughtful feedback and appreciation of our revisions to strengthen our manuscript. We are pleased to know that the additional variants of the cANN model, simulations, and figures have effectively addressed Reviewer #1's concerns, and have subsequently made our manuscript more compelling and appealing to an extensive range of readers in the fields of neuroscience.

There are a few remaining questions. First, the prediction error into the network is equated with the climbing fiber system. Does this predictive error have any properties that would be somewhat like the rather unique climbing fibers and the complex spikes evoked in Purkinje cells? One thought is that the climbing fiber input is a type of non-linearity, and interesting they used a non-linearity to model the recurrent input.

(Regarding "the recurrent input" at the end of this comment, given the context, we interpreted this word as referring to “the climbing fiber input”.)

Our cerebellar ANN (cANN) model is designed to emulate the properties of the climbing fiber system, specifically in its generation of prediction errors. Nonetheless, the model does not emulate the climbing fiber system's non-linearity. The reasoning for this is based on our premise that for cognitive tasks, the linear encoding of prediction errors is adequate (further discussion follows).

In the climbing fiber system, the prediction error is considered to be calculated by subtracting the cerebellar output (the prediction) from the target of prediction (the correct answer). This subtraction happens in the Inferior Olive, which receives excitatory inputs (from outside the cerebellum) that relay the prediction target and inhibitory inputs (from the cerebellar nucleus) that relay the cerebellar prediction. Similarly, our cANN model calculates the prediction error by subtracting the cerebellar output from the prediction target (i.e., correct answer of the next word), thereby replicating the physiological mechanism.

As pointed out by the reviewer, the climbing fiber does exhibit non-linearity. Specifically, the climbing fiber can linearly encode prediction error by increasing its firing rate (Soetedjo, Kojima and Fuchs, 2008). However, it fails to maintain this linearity for the opposite direction of the prediction error, because its firing rate reaches zero, albeit infrequently, due to the low spontaneous firing rate (Ohmae & Medina, 2015). Nonetheless, interestingly, during more complex motor tasks, the rate of change in climbing fiber firing in response to prediction errors is small (Kitazawa et al. 1998). Moreover, the information related to complex movement and its associated errors can still be encoded within a linear range, even when the firing rate of climbing fiber is lessened (Streng, Popa and Ebner, 2017). In this study, given that language processing is a complex cognitive task, we assumed that prediction errors are encoded within the linear dynamic range.

Soetedjo, R., Kojima, Y., & Fuchs, A. F. (2008). Complex spike activity in the oculomotor vermis of the cerebellum: a vectorial error signal for saccade motor learning?. *Journal of neurophysiology*, 100(4), 1949-1966.

Ohmae, S., & Medina, J. F. (2015). Climbing fibers encode a temporal-difference prediction error during cerebellar learning in mice. *Nature neuroscience*, 18(12), 1798-1803.

Kitazawa, S., Kimura, T., & Yin, P. B. (1998). Cerebellar complex spikes encode both destinations and errors in arm movements. *Nature*, 392(6675), 494-497.

Streng, M. L., Popa, L. S., & Ebner, T. J. (2017). Climbing fibers predict movement kinematics and performance errors. *Journal of neurophysiology*, 118(3), 1888-1902.

Second, for the additional ways in which they modified the network in response to my concerns, the results were almost universally positive. While on one level this is good, it also begs the question. Is this a feature of these powerful neural networks, or is this a property of the cerebellar architecture?

The cANN variants we examined largely exhibited language processing capabilities on par with the original cANN, leading Reviewer #1 to question whether such capabilities are inherent in any powerful neural networks or are specific to the cerebellar architecture. To address this, it is important to clarify that while our variants indeed exhibited impressive language processing capabilities, they were not universally effective.

As demonstrated in our simulations, upon removal of the recurrent pathways in cANNs, a notable degradation in language processing was observed. This occurred in both the original non-convergent and the convergent cANNs, suggesting the essential role of recurrent pathways in our cANNs.

Another crucial aspect is the ability to generate a substantial number of candidates for next-word prediction. In the convergent cANN (in Fig. 5), where each module outputs one candidate for the next word, at least ten modules are necessary, and when the number of modules is less than or equal to five, the prediction accuracy significantly drops (see L737-739). This can be attributed to the fact that a sentence can branch into multiple grammatically correct pathways, making a set of five modules insufficient to learn to predict the next word candidates after various branches (see L739-741).

The results of these cANN variant simulations strongly indicate that the impressive performance is not a universal trait across all ANNs, but rather the result of key structural elements within our cANN, which is designed based on cerebellar architecture. These findings

also help us understand which cerebellar circuit features are pivotal for language processing, thereby emphasizing the value of our simulations.

Third, in lines 196-198 the statement is made that the cerebellum must acquire language processing without the cerebral cortex. This does not seem realistic and not certain way this claim is made. Whatever the cerebellum does, it does in concert with many other structures and certainly with the cerebral cortex for language processing. In the Discussion, the authors state that the cerebellum acquires language processing in concert with the cerebral cortex.

We appreciate the reviewer's comment and agree with the point raised. Originally, we stated, "For the cerebellum to assist the neocortex in language acquisition, the cerebellum must acquire language processing on its own without information from the neocortex." (first revision, Lines 195-197). However, the reviewer rightfully indicated an error in this statement, specifically the phrase "without information from the neocortex." We intended to convey that to support the neocortex in language acquisition, the cerebellum must acquire language processing abilities that are yet to be developed by the neocortex. Accordingly, we have revised the statement in the manuscript as follows: "For the cerebellum to assist the neocortex in language acquisition, the cerebellum must acquire language processing **abilities that the neocortex has not yet acquired.**" (L196-198). We thank Reviewer #1 for the thorough peer review and remarking.

Fourth, line 382 comments that cerebellar predictive processing can be on the order of 500 sec. Here the work of Popa et al should be included that found evidence for predictive signals in Purkinje cell simple spike firing for long times including up to 2 seconds (Popa et al, 2012, J Neurosci 32(44):15345-58 and Popa et al, 2017 eNeuro 4).

These are very relevant and important references. We have cited them appropriately in the relevant sections (Lines 110 and 384). We thank Reviewer #1 for pointing us to these papers that reinforce the essential parts of the story of this study.

We appreciate the time and effort Reviewer #1 has taken to review our manuscript in detail and for guiding us toward creating a comprehensive and convincing manuscript. Reviewer #1's expertise and insight have played an invaluable role in this process. We look forward to the possibility of our study contributing to a broader understanding within the field of neuroscience.

Reviewer #2 (Remarks to the Author):

The authors have provided a very detailed response and made substantive changes that have greatly improved the manuscript, leaving no significant concerns.

We are sincerely grateful for Reviewer #2's acknowledgment of the improvements we have made to our manuscript. It was our primary aim to comprehensively address each concern raised, and we are glad to hear that these revisions have been met with the approval of Reviewer #2. We thank Reviewer #2 once again for the time, expertise, and constructive feedback during this process.

REVIEWERS' COMMENTS

Reviewer #1 (Remarks to the Author):

The authors have satisfactorily responded to my additional concerns and I have no further comments or issues with the manuscript. This is an important and novel work that will be of wide interest to the neuroscience community,

REVIEWERS' COMMENTS

Reviewer #1 (Remarks to the Author): The authors have satisfactorily responded to my additional concerns and I have no further comments or issues with the manuscript. This is an important and novel work that will be of wide interest to the neuroscience community,

OUR RESPONSE

We sincerely appreciate Reviewer #1's constructive feedback during the review process. The insightful comments have been instrumental in enhancing our manuscript, significantly broadening its appeal to a diverse audience. We are eager to see our research contribute meaningfully to the neuroscience field.